# A low-power vertical dual-gate neuro-transistor with short-term memory for high energy-efficient neuromorphic computing

Han Xu[1,2,3,4], Dashan Shang [1,2,3] ✉, Qing Luo [1,2,3], Junjie An[1,2], Yue Li[1,2,3], Shuyu Wu[1,2,3], Zhihong Yao[1,2], Woyu Zhang[1,2,3], Xiaoxin Xu[1,2,3], Chunmeng Dou[1,2,3], Hao Jiang[5], Liyang Pan[4], Xumeng Zhang [5], Ming Wang [5], Zhongrui Wang [6], Jianshi Tang [4] ✉, Qi Liu[1,2,5] ✉ & Ming Liu [1,2,5]

Neuromorphic computing aims to emulate the computing processes of the brain by replicating the functions of biological neural networks using electronic counterparts. One promising approach is dendritic computing, which takes inspiration from the multi-dendritic branch structure of neurons to enhance the processing capability of artificial neural networks. While there has been a recent surge of interest in implementing dendritic computing using emerging devices, achieving artificial dendrites with throughputs and energy efficiency comparable to those of the human brain has proven challenging. In this study, we report on the development of a compact and low-power neurotransistor based on a vertical dual-gate electrolyte-gated transistor (EGT) with short-term memory characteristics, a 30 nm channel length, a record-low read power of ~3.16 fW and a biology-comparable read energy of ~30 fJ. Leveraging this neurotransistor, we demonstrate dendrite integration as well as digital and analog dendritic computing for coincidence detection. We also showcase the potential of neurotransistors in realizing advanced brain-like functions by developing a hardware neural network and demonstrating bio-inspired sound localization. Our results suggest that the neurotransistor-based approach may pave the way for next-generation neuromorphic computing with energy efficiency on par with those of the brain.

The human brain possesses unparalleled cognitive capabilities, occupies a small footprint (roughly the size of a football; ~1200–1700 cm³, depending on individuals), and yet consumes very little power (approximately 20 W). Neuromorphic computing seeks to emulate the structure and functions of the human brain using electronic counterparts, thus replicating its area- and energy-efficiency[1–5]. The human

brain is a biological neural network (BioNN) composed of neurons, dendrites, and synapses, which has inspired the development of artificial neural networks (ANNs) that have had transformative impacts on computer vision, speech recognition[6], and bioinformatics[7]. Nowadays, ANNs predominantly operate on digital hardware, which is not improving at an exponential pace anymore due to the slowdown of

[1]State Key Lab of Fabrication Technologies for Integrated Circuits, Institute of Microelectronics, Chinese Academy of Sciences, Beijing 100049, China. [2]Key Laboratory of Microelectronics Devices and Integrated Technology, Institute of Microelectronics, Chinese Academy of Sciences, Beijing 100049, China. [3]University of Chinese Academy of Sciences, Beijing 100049, China. [4]School of Integrated Circuits, Beijing National Research Center for Information Science and Technology (BNRist), Tsinghua University, Beijing, China. [5]Frontier Institute of Chip and System, Fudan University, Shanghai 200433, China. [6]Department of Electrical and Electronic Engineering, The University of Hong Kong, Hong Kong 999077, Hong Kong. ✉e-mail: shangdashan@ime.ac.cn; jtang@tsinghua.edu.cn; qi_liu@fudan.edu.cn

Moore's Law, limiting the ability to make increasingly complex ANNs without increasing compute times. Moreover, the von Neumann bottleneck, an inherent limitation of digital hardware, further hinders the execution efficiency of ANNs. Emerging non-volatile (or long-term memory, LTM) memories and their crossbar integration have shown advantages as hardware synapses[8–15]. Nevertheless, the performance of ANNs still lags behind that of the brain, because the brain's computing power largely relies on the complex chemical cascades that occur within neuron cells[16]. This necessitates a more faithful emulation of the structure and dynamic behaviors of neurons to unleash the throughput and efficiency of neuromorphic computing[17,18].

To address this grand challenge, there is a tremendous effort underway to devise new building blocks for neuromorphic systems. For instance, basic neural functions, such as integration-and-fire, have been implemented on short-term memory (STM) devices[19–23]. In BioNN, a neuron interacts with over 1000 adjacent neurons through its dendrites, processing massive spatiotemporal signals through its dendrite[24,25]. This dendritic computing paradigm endows the neuron with complex behaviors and computing power[26,27]. For example, there is evidence that dendritic computing is responsible for the sound localization capabilities of the brain[28]. Therefore, to emulate neural computing with new fidelity, there has been a recent surge of interest in devices that have both STM and multi-dendritic structures[9,29].

Neurotransistors are among the devices that have demonstrated important computational functions of neurons[19,20,22,23]. For example, a neurotransistor derived from a proton-conducting graphene oxide electric-double-layer transistor (EDLT), with a footprint of 80 μm × 240 μm (channel length × channel width) and a read power of tens of nW, mimicked the dendrite integration, orientation tuning, and gain control of neurons[19]. Moreover, a multi-terminal neurotransistor that can emulate the dendritic discrimination of neurons for different spatiotemporal signals was also developed from EDLT, which employs coplanar source-drain electrodes and gate and has a closest gate-to-channel distance of 565 μm[20]. The energy consumption of this neurotransistor in response to a single gate voltage pulse is approximately 1 nJ. However, these three-terminal or multi-terminal devices, which emulate how neurons compute, suffer from a large footprint, high energy consumption, and poor scalability, which significantly undermine their advantages in area and energy efficiency.

In this study, we employed material and structure engineering to develop a vertical EGT (V-EGT) based neurotransistor that simultaneously exhibits STM, an ultra-short channel (30 nm), low energy consumption (read power ~3.16 fW, read energy ~30 fJ), and dual gates, making it an ideal choice for hardware implementation of dendritic computing, such as dendrite integration, digital and analog coincidence detection. Moreover, by integrating neurotransistors into a prototypical hardware BioNN, we demonstrated the ability to emulate the sound localization function of the brain, including sound azimuth and distance recognition. Our small size, low power neurotransistor, and the proof-of-concept neural network for sound localization highlight their potential in developing neuromorphic systems that can achieve the energy-efficiency of the brain.

## Results

Figure 1a depicts the schematic of the fabrication process flow of V-EGT, which comprises three main steps. The first step (i) involves electrode/spacer/electrode/spacer stack deposition, the second step (ii) is one-step etching to the substrate, and the third step (iii) is channel/electrolyte/gate stack deposition. Thanks to the exposed vertical sidewall, constructing multi-gate V-EGTs is easily achievable by depositing multiple channel/electrolyte/gate stacks side-by-side along the vertical sidewall (the right half of Fig. 1a iii), which is a challenging task for planar EGTs. Furthermore, this V-EGT fabrication process enables an ultra-short-channel without significantly increasing the fabrication complexity compared to planar EGT (Supplementary

Fig. 1). Our V-EGTs are designed to be compatible with CMOS processes, making them particularly suitable for large-scale, low-cost fabrication (Supplementary Fig. 2). This is due to the absence of organic/2D channels and liquid/gel-like electrolytes.

The power consumption bottleneck in current neurotransistors is caused by the relatively high conductivity of semiconductor channel materials (such as p-Si[30], ITO[31], IZO[32], IGZO[33], ZnO[34], and In$_2$O$_3$[35]) or lithium-ion battery electrode materials (such as LiCoO$_2$[36,37] and Li$_x$TiO$_2$[38]). In our previous work, we demonstrated that the simple binary metal oxide Nb$_2$O$_5$[39,40] is a promising solution to meet the requirement of high channel resistivity. Additionally, the stable thermodynamic properties of Nb$_2$O$_5$ also contribute to stable and reliable electrical operations[41]. It should be noted that although VO$_2$[42], SrCoO$_x$[43], and SmNiO$_3$[44] meet the high channel resistivity requirement, they are less favorable than Nb$_2$O$_5$ because VO$_2$ is toxic, and SrCoO$_x$ and SmNiO$_3$ are complex in their compositional elements. To achieve all-solid-state neurotransistors, Li$_x$SiO$_2$ was utilized as the solid electrolyte, which can also be conveniently deposited by physical vapor deposition (PVD). Although there exists an incompatibility issue between lithium electrolyte materials and CMOS technology, the industry is actively exploring solutions to this challenge[45].

The device structure was examined using cross-sectional transmission electron microscopy (TEM), confirming the vertical structure of our device (Fig. 1b). In contrast to the single-layer vertical device depicted in Fig. 1a, a total of four layers of electrode/spacer bilayer were deposited in this work, essentially creating a bilayer V-EGT (Supplementary Fig. 3). The spacer and electrode have thicknesses of 30 and 20 nm, respectively, resulting in a minimum channel length of 30 nm. This is a key feature that highlights the main difference between V-EGTs and planar EGTs. Specifically, V-EGTs address the issue of device scaling that has hindered planar EGTs by ingeniously defining the channel length of EGTs with the thickness of thin films. Furthermore, the vertical stackability of V-EGTs, made possible by the easy cycling deposition of electrode and spacer, endows them with the merit of variable channel length, which will be discussed in detail later. The multi-gate nature of V-EGTs also provides an opportunity to implement various functionalities that exist in the brain, further enhancing the superiority of V-EGTs over planar EGTs.

Elemental mapping (Fig. 1c) corresponding to Fig. 1b confirms the presence and spatial distribution of the Nb$_2$O$_5$ channel, Li$_x$SiO$_2$ electrolyte, TiN source-drain, Ti/Au gate, and SiO$_2$ spacer, revealing the sharp boundaries between them (Fig. 1c and Supplementary Fig. 4). The zoomed-in high-resolution cross-sectional TEM image illustrates the vertical structure of a single V-EGT (Fig. 1d). Unlike planar EGTs with exposed source/drain electrodes, the 3D stacked TiN electrodes of different layers in the TEM are interfaced with an array of non-overlapping probing pads for electrical testing. In addition to minimizing the EGT channel length by adopting a vertical structure, the channel width of the EGT was also scaled down as much as possible, resulting in the smallest EGT (channel length × channel width = 30 nm × 2 μm = 0.06 μm$^2$). An elemental line scan along the V-EGT channel/electrolyte/gate stack direction (dashed line in Fig. 1d) indicates that our V-EGT has a channel thickness of less than 10 nm and an electrolyte thickness of less than 20 nm (Fig. 1f). Therefore, our V-EGT not only features a significantly reduced horizontal dimension (down to 30 nm) but also a vertical dimension (down to 20 nm).

To highlight the advantages of our V-EGT, we have compared it with recently reported V-EGTs (Supplementary Table 1)[46–50]. Our V-EGTs demonstrate three key advances. First, we have optimized the etching process of the vertical sidewalls, resulting in steeper and smoother sidewalls (Fig. 1d). Second, we have demonstrated dual-gate V-EGTs. Lastly and most importantly, our V-EGTs employ different types of electrolytes, enabling STM electrical characteristics. The compact dual-gate V-EGT, with STM and ultra-low power, is an ideal building block for high area- and energy-efficient neuromorphic systems.

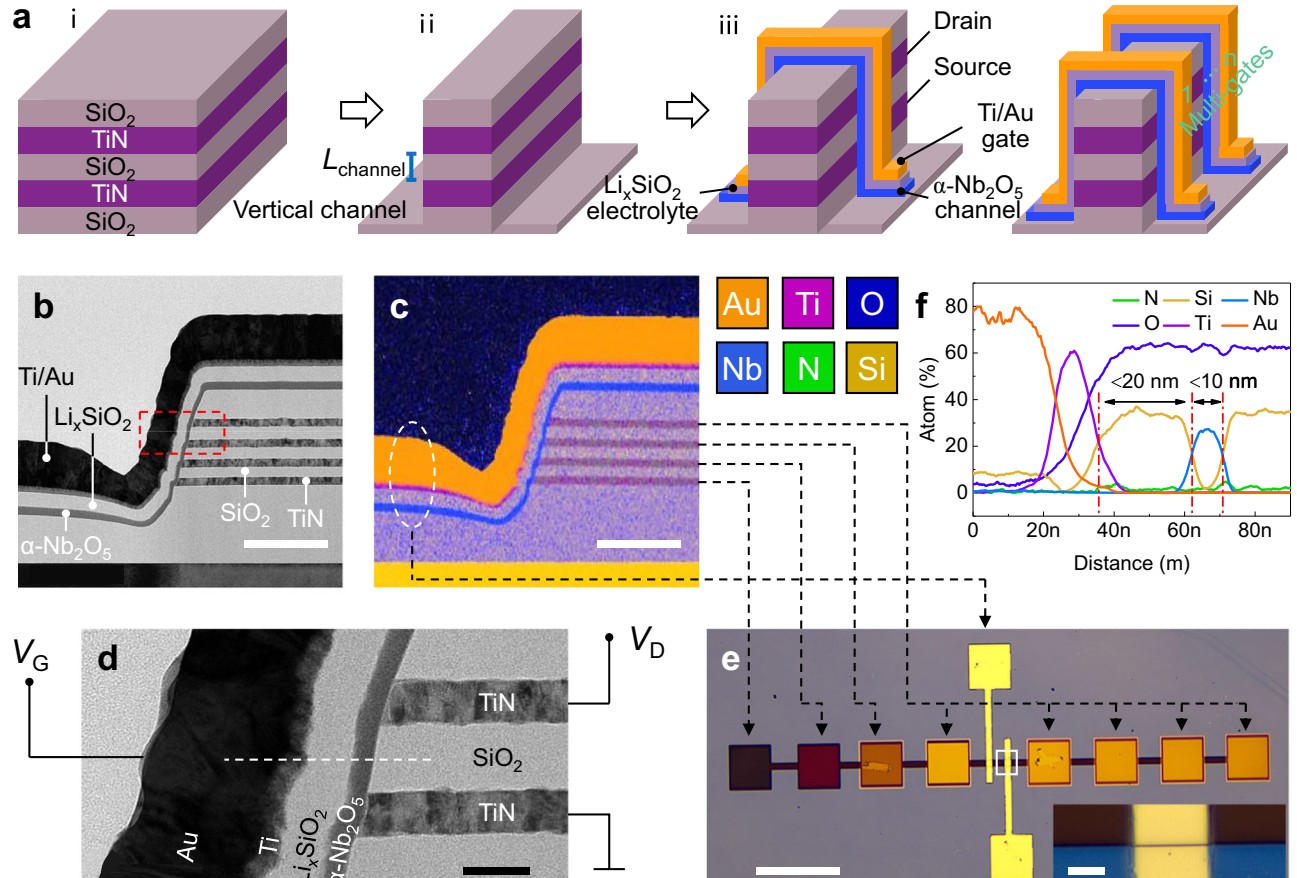

**Fig. 1 | Device structure of V-EGT. a** Device fabrication process flow of V-EGT. The purple color represents the source-drain electrodes, gray represents the substrate, spacer, or electrolyte, blue represents the channel, and yellow represents the gate. **b** Cross-sectional TEM image of a V-EGT. Each V-EGT has a two-layer vertical structure without sharing any source and drain electrodes, with four layers of source-drain electrodes along the vertical direction. **c** Energy-dispersive X-ray spectroscopy (EDX) element mapping corresponding to (**b**). The scale bars in (**b** and **c**) are 200 nm. **d** Magnified TEM image of a single V-EGT highlighted by the red box in (**b**) (scale bar is 30 nm). The elemental composition of each layer of the V-EGT and the typical operation voltage are labeled. **e** Optical microscopy (OM) image of a two-layer V-EGT. The inset shows the zoomed coverage of the gate at the vertical sidewall. The one-to-one correspondence between the electrodes in OM and the electrodes of each layer in TEM is given. The scale bars in (**e**) and its inset are 100 and 5 μm, respectively. **f** Elemental line scan along the V-EGT channel/electrolyte/gate stack direction (dashed line in **d**).

Like planar EGTs, V-EGTs are also operated by biasing both the gate and drain while grounding the source (Fig. 2a, b). Furthermore, both V-EGTs and planar EGTs rely on the migration of electrolyte ions, specifically the hybrid electric double layer and ion intercalation/deintercalation mechanism. This unique feature of EGT (Supplementary Fig. 5)[51] is also the origin of the observed coexistence of STM and LTM in our case, with the former dominating (Fig. 2c and Supplementary Fig. 6). Regarding the obvious STM of our V-EGT, Li's analysis on the correlation between the retention and device size of Li-ion-based EGTs (Li-EGTs) can be applied. According to Li's analysis, the retention (STM) of Li-EGTs decreases (increases) with decreasing channel area[52]. Specifically, the retention of EGT depends on the discharge speed of EGT in the gate-source circuit after experiencing a gate pulse ($\tau_{RC} = R \times C$) from a circuit perspective. Here, R comprises the EGT electrolyte resistance and the external resistance in series with EGT, and C contains the gate and channel capacitance of EGT (Fig. 2d and Supplementary Fig. 7)[53]. For Li-EGT that is in series with an electronic switch (1S1E), the R in the discharge circuit is mainly determined by the OFF state resistance of the electronic switch, which does not change with the size of Li-EGT. However, the C of the discharge circuit decreases with the reduction of Li-EGT size. Therefore, the retention of 1S1E decreases with decreasing channel area (Fig. 2e). Although this situation applies to the retention of 1S1E or Li-EGT in open circuits, it remains true for our case or Li-EGT in short circuits. This is because the

short-circuit condition has a smaller discharge load resistance than the open-circuit condition (which only contains the electrolyte resistance of Li-EGT and no external resistance), resulting in a faster discharge speed and more obvious STM.

Benefiting from the stackability of our V-EGTs, the channel length can be flexibly tuned. For instance, EGTs with channel lengths of 30 nm (using the first and second layers of TiN electrodes), 80 nm (using the first and third layers of TiN electrodes), and 130 nm (using the first and fourth layers of TiN electrodes) were demonstrated by selecting different source-drain electrodes. All of them exhibited typical electrolyte-gated behaviors (Supplementary Fig. 8). We also examined the symmetry of the source-drain electrodes and observed that the transfer characteristics of different source-drain electrode combinations were consistent, without any impact from vertical sidewall tilt (Supplementary Fig. 9).

Figure 2f and g display the current-time response of V-EGT under pulses of varying amplitudes and widths. Regardless of the amplitude or width of the gate pulse, the channel current of V-EGT decays over time after the pulse excitation, indicating the presence of STM. Furthermore, as the pulse amplitude or width increases, the device exhibits a gradually increasing residual channel current. We quantified the change in conductance as $((G_1 - G_0)/G_0) \times 100\%$, where $G_0$ represents the channel conductance before the pulse application and $G_1$ is either the channel conductance right after the pulse application or after 30 s,

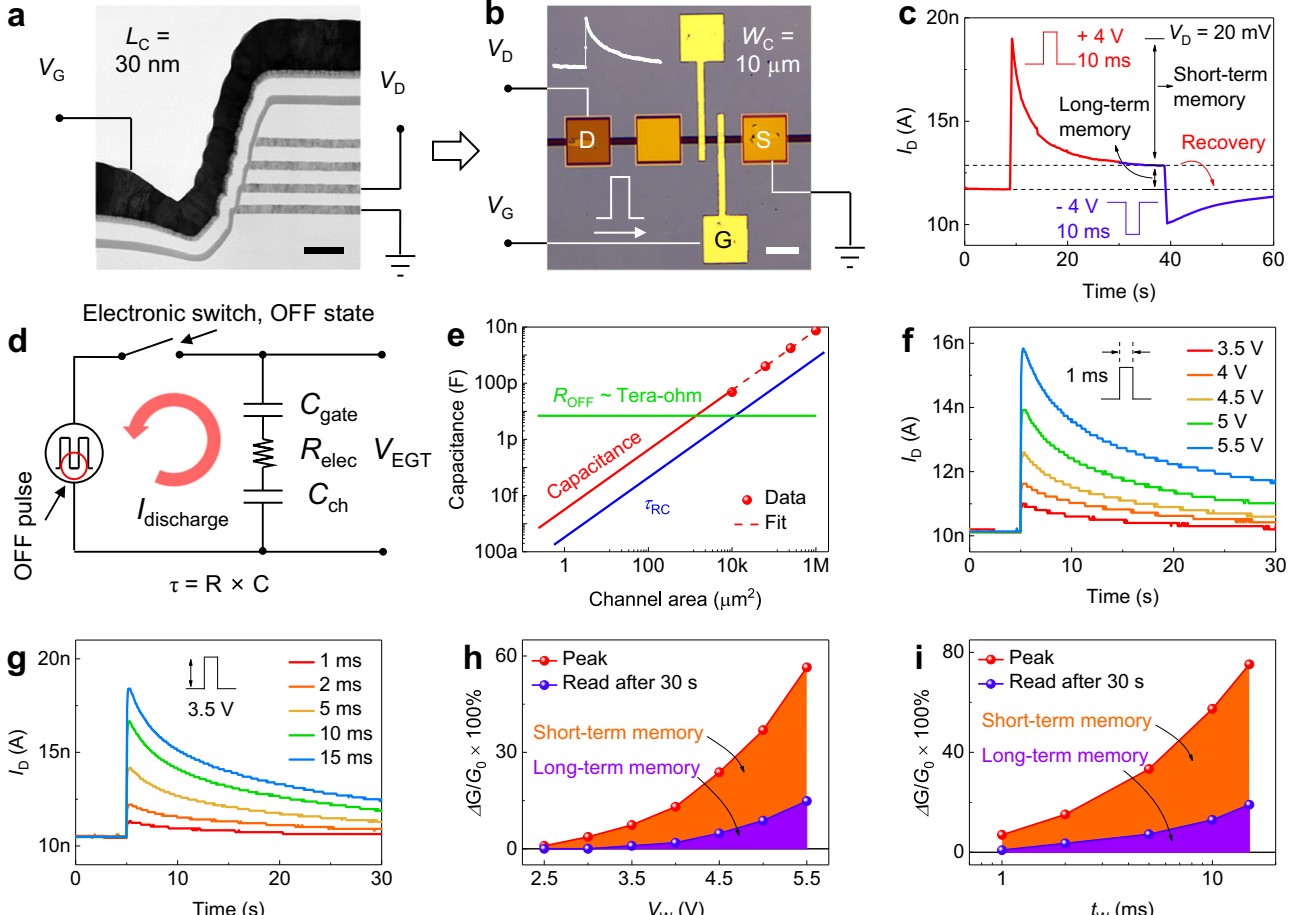

**Fig. 2 | Electrical properties of V-EGT. a, b** Schematic diagram of the electrical testing setup for V-EGT. The scale bars in (**a** and **b**) are 100 nm and 50 μm, respectively. **c** Current response of V-EGT under voltage pulses, showing hybrid short-term memory (STM) and long-term memory (LTM) with the former dominated. **d** Equivalent circuit diagram of the EGT gate-source circuit. In the intervals between gate pulses, EGT discharges along the original charging path, as indicated by the red arrow. The discharge time constant ($\tau_{RC} = R \times C$) is an indicator of EGT retention. **e** The discharge time constant of Li-ion based EGTs in series with electronic switches decreased with decreasing channel area. **f, g** Current-time response of V-EGTs under different voltage pulse amplitudes (**f**) and widths (**g**). **h, i** Analysis for the pulse intensity (amplitude (**h**) and width (**i**)) dependence of STM and LTM of V-EGT.

corresponding to the STM and LTM, respectively. Figure 2h and i show the conductance change ratios of V-EGT as a function of the write voltage pulse amplitude ($V_w$) and width ($t_w$), respectively. When $V_w \leq 3$ V ($t_w = 1$ ms) or $t_w < 1$ ms ($V_w = 3.5$ V), the LTM conductance change ratio is 0. As the amplitude and width of the write voltage pulse increase, the LTM becomes increasingly obvious. To enable the V-EGT to work as an artificial neuron, we engineered the amplitude and width of the gate voltage pulses to produce nearly complete STM.

The energy consumption of EGTs during write and read operations is a crucial factor to consider. Since the gate-source path of EGTs is capacitive in nature, the read operation dominates the energy consumption[54]. While the read operation of EGTs typically involves applying a DC voltage to the drain, resulting in a finite read energy consumption being unavailable, the read power ($P_R = V_R \times I_R$) is a more meaningful metric to evaluate EGT performance.

Planar EGTs have a larger channel length (Supplementary Fig. 10), leading to a higher read voltage and, consequently, higher read power consumption. Conversely, V-EGTs can significantly reduce their read voltage and read power consumption because the channel length is no longer limited by lithography technology but instead by material thickness (Fig. 3a). With the channel thickness, width, and current remaining constant, $V_D$ is calculated to decrease linearly with the decrease in channel length (Fig. 3b). Therefore, V-EGTs can achieve a substantial reduction in their read voltage, and subsequently, their

read power consumption. In this study, we demonstrated an ultra-low read voltage of 0.1 mV for a 30 nm channel length V-EGT (Fig. 3c). Moreover, the channel current was found to be obviously larger than the gate leakage at $V_D = 0.1$ mV (-150 pA Vs. -50 pA), thereby validating the effectiveness of the 0.1 mV read voltage (Fig. 3d). We also compared the ultra-low read voltage of our V-EGT with other EGTs (Supplementary Fig. 11). While some EGTs have achieved read voltages of 0.1 or even 0.01 mV, they consist of organic channel and liquid electrolyte, posing difficulties in large-scale manufacturing. In contrast, our V-EGTs not only offer easy and cost-effective fabrication at a large-scale but also possess the lowest read voltage (0.1 mV) among all inorganic all-solid-state EGTs[54–63]. Furthermore, reducing the gate width of the EGT can further decrease the read current of the device, thereby reducing the read power consumption (Fig. 3e and Supplementary Fig. 12).

To calculate the read power of V-EGTs, we characterized the current-time response of our devices upon voltage pulses at different read voltages. The curve corresponding to the 1 mV read voltage is clean and free of fluctuation (Fig. 3f). Furthermore, we evaluated the read power of the device (Fig. 3g). Thanks to the small read voltage and the resulting small read current, we achieved a minimum measured read power of 110 fW ($V_w$ (4 V, 10 ms), $V_r$ (0.5 mV)). Moreover, the use of narrower write voltage pulses and smaller read voltages leads to a read power of the V-EGT in the fW-level, specifically ~3.16 fW

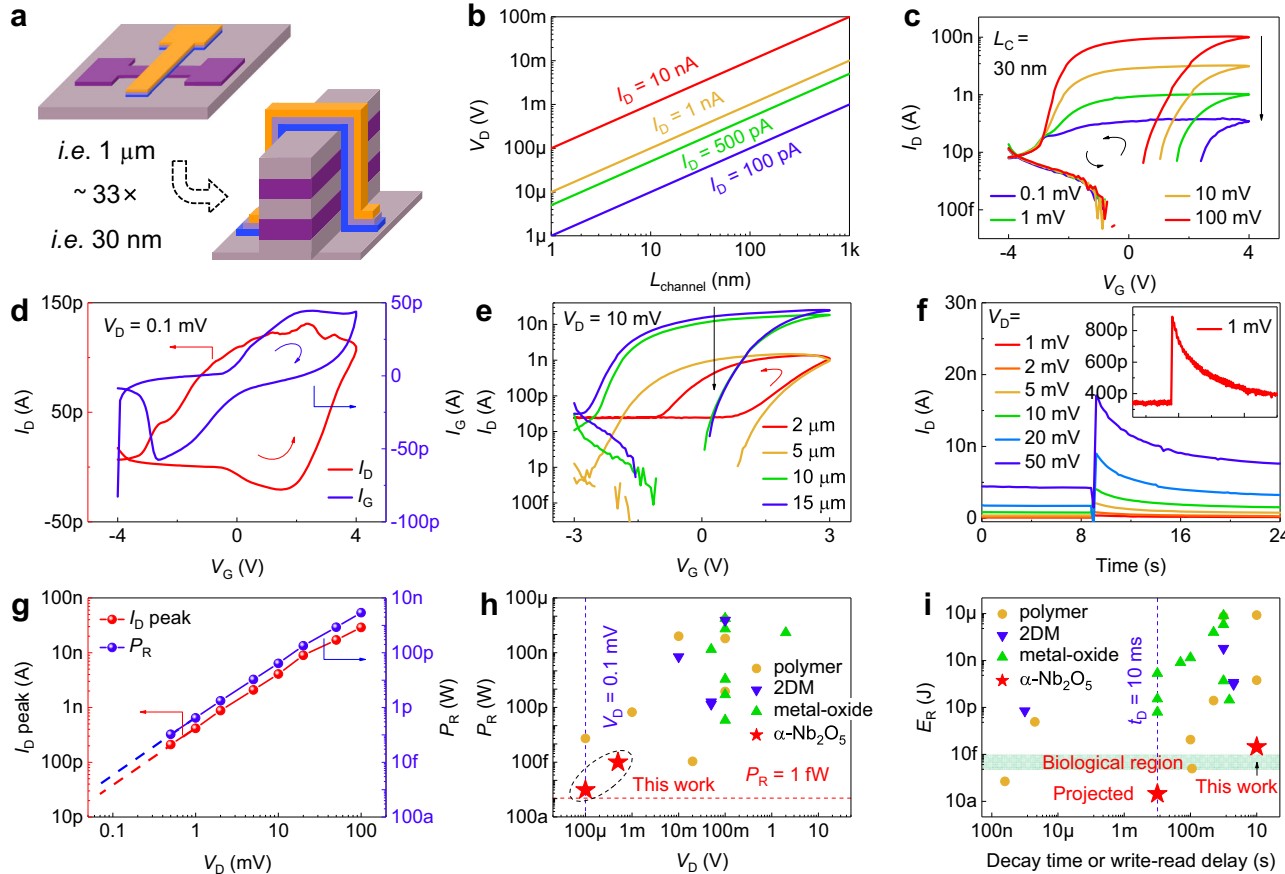

**Fig. 3 | Read voltage, power, and energy consumption of V-EGT. a** The channel length of planar EGT is limited by lithography technology, while that of V-EGT is limited by material thickness, enabling the latter to significantly reduce the channel length. **b** Relationship between $V_D$ and $L_{channel}$ with fixed channel thickness, width, and channel current. Situations with different channel currents are shown. **c** Transfer characteristics of the V-EGT ($L_{channel} = 30$ nm) at different read voltages. **d** Transfer characteristic and gate leakage of the V-EGT ($L_{channel} = 30$ nm) at $V_D = 0.1$ mV. **e** Transfer characteristics of V-EGTs with different gate widths.

**f** Channel current-time response of the V-EGT at different read voltages. The inset zooms in on the curve corresponding to 1 mV read voltage. The gate pulse condition is $V_W = 4$ V, $t_W = 10$ ms. **g** The relationship between the peak read current and read power of the V-EGT versus the read voltage, according to data shown in (**f**). **h** Comparison among various EGTs in terms of read power. The material type of the used channel has been identified. 2DM refers to 2D materials. **i** Comparison among various EGTs in terms of read energy consumption. The material type of the used channel has been identified. 2DM refers to 2D materials.

(Supplementary Fig. 13). To the best of our knowledge, such a read power is the lowest among various EGTs (Fig. 3h). In addition to analyzing the read power, we have also computed and compared the read energy consumption of our devices with that of others. Initially, we formulated an equation to calculate the energy of both the STM and LTM EGTs. For STM EGTs, we estimated the time for calculating the read energy consumption by measuring the time required for the decay of STM EGTs from the maximum channel current to the initial current after undergoing a pulse. To ensure a fair comparison with STM EGTs, we used the write-read delay of LTM EGTs as the time for calculating read energy consumption (refer to Supplementary Note 1 for further details). We then computed and compared the read energy consumption of various EGTs (Supplementary Table 2 and Fig. 3i). Despite the relatively long decay time of our V-EGT (~10 s), we still achieved a read energy consumption of approximately 30 fJ due to the ultra-low read power, which is comparable to the energy consumption of biology (1–10 fJ). By expediting the decay process of V-EGT (refer to Supplementary Note 2 for further details), it is possible to decrease the decay time and hence the read energy consumption of the V-EGT. With a projected decay time of 10 ms, the V-EGT could potentially achieve a read energy of approximately 30 aJ (Fig. 3i). Apart from the read energy, write energy is also an important aspect of EGT's energy profile, although it can be neglected due to the minimal gate leakage current. The key challenge in write energy estimation is to measure

accurately the write current. By assuming an average gate leakage current of 50 pA (the maximum value observed in Fig. 3d), based on Supplementary Fig. S13c, we have estimated the write energy of V-EGT (275 fJ = 50 pA × 5.5 V × 1 ms) and compared it with other literatures (Supplementary Table 3).

In addition to low energy consumption, the vertical structure of V-EGTs also offers inherent advantages in device density. Compared with planar EGTs, V-EGTs boast a 2.5-fold increase in device area ($4F^2$ vs. >$10F^2$, where F refers to the feature size) (see Supplementary Note 3 for more discussion on the calculation of V-EGT device footprint in array configuration)[49]. Furthermore, when the vertical stackability of V-EGTs is exploited in the future, their dimensions could be further reduced by a factor of N, where N represents the number of V-EGT layers[64]. It is important to note that the significantly decreased channel length of V-EGTs will not only result in a substantial reduction in the read voltage, but also facilitate the potential enhancement of the write voltage and speed of the EGT. Through an analysis of the current composition of EGT and the interdependence between them[65], it was discovered that the electrolyte thickness has the ability to synergistically scale with the channel length of EGT (Supplementary Fig. 14a–c). Thanks to the significantly reduced channel length of V-EGT, the electrolyte thickness of V-EGT is thinner than that of planar EGTs (~20 nm vs. ~50 nm for planar EGTs), which consequently improves the write voltage and speed (~5-fold increase compared to planar EGTs)

(Supplementary Fig. 14d, e). Ultimately, the overall scaling down of V-EGT (including the channel length and width, as well as the electrolyte thickness) results in concurrent low write and read voltages (Supplementary Fig. 15).

The compact size and multi-gate structure, along with the features of STM and low energy consumption, make our V-EGTs highly suitable for use as high-density and low-power artificial neurons. In this regard, we first demonstrate that V-EGTs and biological neurons exhibit similar relationships that govern the input-output signals by using a single dual-gate V-EGT (Fig. 4a). In this setup, with the gate of a dual-gate V-EGT being considered as the dendrite of a neuron and the drain of a dual-gate V-EGT as the axon of a neuron, our device can mimic signal propagation pathways in the brain. Furthermore, the short-term response of the channel current to gate voltage pulses and the integration function of the drain current to multiple gate voltage pulses reflect the short-term information processing capability of the neuron and the dendrite's spatial integration of multiple input signals. Figure 4b illustrates the electrical testing setup for the dual-gate V-EGT, which is used to characterize the artificial neuron properties of our device. Firstly, we examined the dendrite integration of our dual-gate V-EGT neurons (Fig. 4c). Since the channel current controlled by each gate is independent, the resulting dendrite integration is linear, i.e., $I_{ds} = I_{ds1} + I_{ds2}$. It should be noted that although the dendrite integration of our dual-gate V-EGT artificial neurons lacks the non-linearity observed in BioNNs, this linear integration mechanism is well-suited for scenarios requiring precise computations, such as logical operations. In biology, different dendrites of a neuron form distinct synaptic weights with neighboring neurons. However, for simplicity, we assume here that each dendrite of a neuron shares the same signal-receiving capacity. Correspondingly, each gate of the dual-gate V-EGT should also have equal ability to regulate the channel current. To verify this, we investigated the consistency between different gates. Owing to the highly uniform etching and thin-film deposition processes, the electrical properties of different gates in the dual-gate V-EGT exhibited good consistency when each gate was controlled individually (Supplementary Fig. 16). Note that, the dual-gate V-EGT devices are not exact the same to the biological neurons. The different gate stacks of the devices have the same synaptic weight, while different dendrites of biological neurons inherently have different synaptic weights. This characteristic of biological neurons can be achieved indirectly by

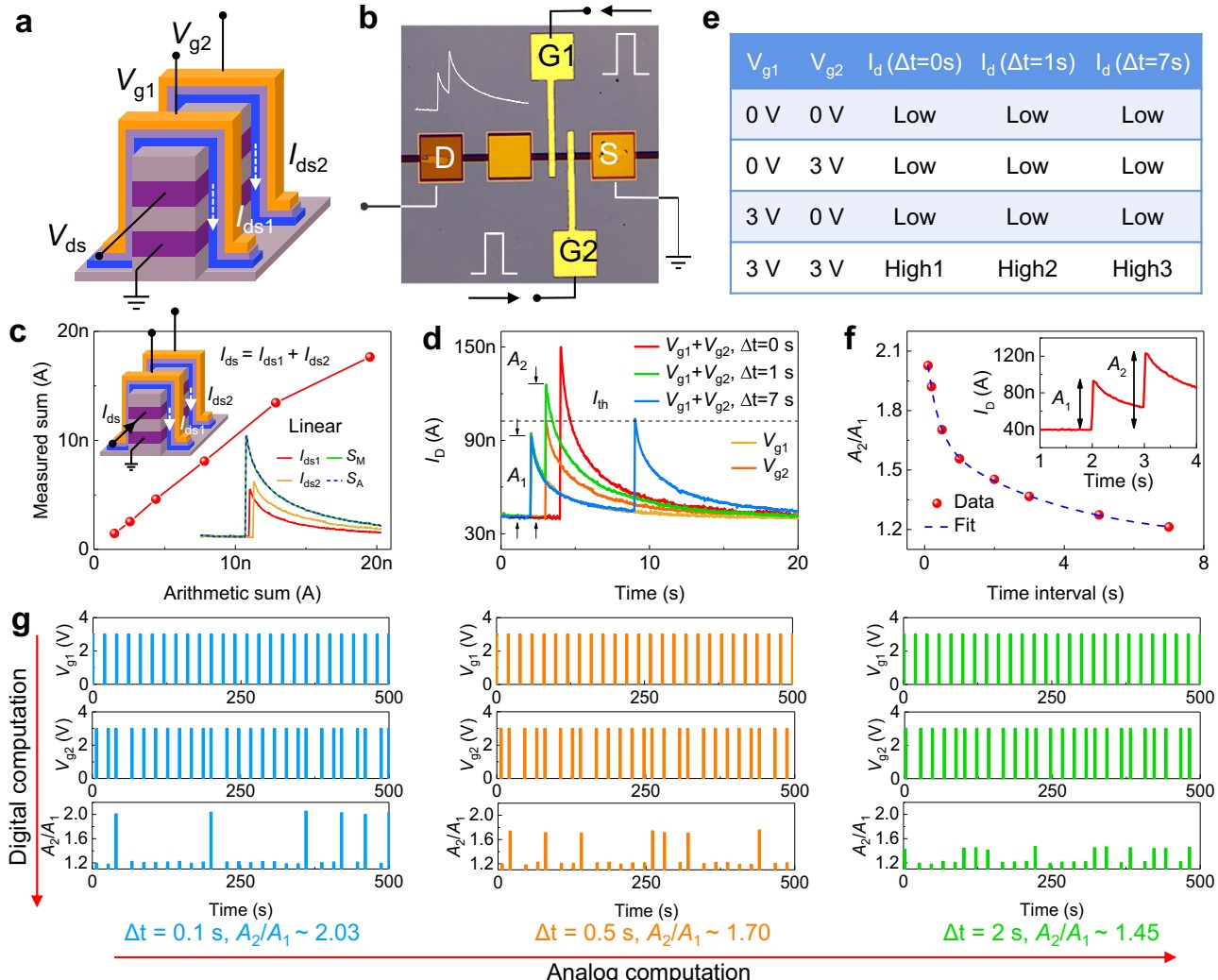

**Fig. 4 | Dendritic computing functions of artificial neurons based on dual-gate V-EGTs. a** Schematic diagram of the dual-gate V-EGT-based artificial neuron. **b** Testing setup for the artificial neuron based on dual-gate V-EGTs. **c** Relationship between the arithmetic sum of the output and the measured sum of the output, showing the dendrite integration function of the artificial neuron. **d** Channel current-time response curves when the two gates act individually or simultaneously. **e** Dual-gate V-EGT realized the AND logic and coincidence detection in dendritic computing by introducing a current threshold $I_{th}$ (the peak channel current at $\Delta t = 0$ is greater than that at $\Delta t \neq 0$). **f** Dual-gate paired pulse facilitation (PPF) of the dual-gate V-EGT. The inset shows an example waveform of dual-gate PPF. **g** Digital and analog computing for coincidence detection of neurons based on dual-gate V-EGTs.

applying different pulses to each gate of the dual-gate V-EGT. (Supplementary Fig. S17). One possible way to achieve the synaptic weights with inherent difference is separately depositing the different channel/electrolyte/gate stacks of the dual-gate V-EGT with different material dimensions.

In addition to realizing the dendrite integration function of neurons, our dual-gate V-EGT can also perform AND logic by utilizing the difference between the drain current when the two gates act simultaneously and separately. This functionality enables the realization of the coincidence detection function of neurons. To achieve this, a current threshold $I_{th}$ is set, and the drain current is only larger than $I_{th}$ when both gates act simultaneously (Fig. 4d). Vice versa, the current is smaller when the gates act separately, which implements the AND logic, as illustrated in Fig. 4e. Interestingly, the peak channel current varies with the time interval between voltage pulses applied to gate 1 and 2, gradually decreasing with the increase of the time interval. Here, we use high1, high2, and high3 to represent the peak of channel currents corresponding to $\Delta t = 0$, 1, and 7 s, respectively, where high1>high2>high3. This characteristic can be utilized for coincidence detection by biological neurons in response to two events. For biological neurons, receiving two input signals simultaneously results in the largest output signal, while receiving them separately results in a weaker output signal. In order to improve the coincidence detection function of our dual-gate V-EGT artificial neurons and apply it to sound distance detection (which will be discussed later), we have expanded the definition of neuronal coincidence detection by adopting a new physical quantity to benchmark the degree of coincidence of two events (Fig. 4f). Specifically, we define the interval between two events as $\Delta t$, where $\Delta t$ can be any real number. With the dual-gate paired-pulse facilitation (PPF) characteristics of our device, coincidence detection can be achieved in other situations (e.g. $\Delta t = 0.1$, 0.5, and 2 s). To differentiate it from the PPF of a single-gate EGT, we refer to the PPF of dual-gate V-EGT as dual-gate PPF, and an example waveform of dual-gate PPF is shown in the inset. The intensity of such coincidence detection can change continuously (non-linear decay) (Fig. 4f), which is crucial for sound distance detection.

To better demonstrate the coincidence detection capabilities of our device for sound distance detection, we present the $A_2/A_1$ values obtained from a series of input pulses with different time intervals ($\Delta t = 0.1$, 0.5, 2, and 7 s) in Fig. 4g. By defining a coincidence threshold of $\Delta t = 0.1$, 0.5, and 2 s respectively, and a non-coincidence threshold of $\Delta t = 7$ s, we were able to observe coincidence detection similar to that observed in biological neurons, as indicated by the vertical trend in Fig. 4g. Specifically, we observed a significantly larger coincidence degree at $\Delta t = 0.1$, 0.5, or 2 s compared to that at $\Delta t = 7$ s, which is indicative of digital coincidence detection. In addition, by varying the criteria of coincidence detection from $\Delta t = 0.1$ s to $\Delta t = 0.5$ s and $\Delta t = 2$ s along the horizontal direction of Fig. 4g, we observed a gradual decrease in the $A_2/A_1$ coincidence degree for the corresponding coincidence detection, which is a manifestation of analog computing-based coincidence detection (refer to Supplementary Note 4 for further details).

The neuronal properties of our device facilitate the reemergence of complex brain functions. Sound localization serves as an example. By defining the front of the human eye as the reference direction, the angle between the propagation direction of sound and the normal is referred to as the azimuth of sound. When sound is emitted, the presence of the azimuth generates a sound path difference, causing the left and right ears to perceive the sound at different times (interaural time difference, ITD) (Fig. 5a). As ITD and sound path difference have a one-to-one correspondence, the brain can determine the sound azimuth through ITD (Fig. 5b).

To replicate the sound localization abilities of the brain accurately, it is essential to predict the distance of the sound source in addition to recognizing the sound azimuth. For instance, consider the

human brain's ability to estimate the distance of a truck driving on the road ahead. As the intensity of sound decreases proportionally with the square of the distance, the brain can determine the distance of the truck based on the intensity of sound it receives (Fig. 5c). When the truck is to the listener's left and moves from left to right, the listener perceives a gradual increase in the volume of the truck's honk. As the truck moves towards the right and away from the listener, the sound of the honk gradually diminishes (Fig. 5d).

To distinguish the ITD of sound and identify its azimuth and distance, we constructed a neural network (Fig. 5e). The blue spheres in the figure represent neurons that sense sound in the left and right ears, respectively. These neurons transmit sound information to post-sound information processing neurons (green or yellow spheres) through axons (blue lines), synapses (blue nodes), and dendrites (green or yellow lines). The sound signal is assumed to propagate through the network in the form of spikes, where the amplitude of spikes is modulated by synaptic weight. By configuring the synaptic weight matrix between the pre-neurons and two green post-neurons into a diagonal matrix, the outputs of the two post-neurons can be different. The difference between the outputs of the two neurons varies with the ITD and is symmetric with respect to the ITD (specifically, the output of neuron 1 (3) at ITD = -t is the same as the output of neuron 3 (1) at ITD = t). The network can recognize sound azimuth accordingly (see Supplementary Note 5 for a more detailed working principle analysis). The post-neuron situated in the center of the three post-neurons exhibits equivalent connection strength to the two neurons that sense sound in the left and right ears, as shown in Fig. 4 while introducing the coincidence detection function of neurons. As the distance between the truck and the listener increases, the intensity of the honking sound heard by the listener diminishes. This precisely corresponds to the fact that a larger ITD results in a smaller coincidence degree $A_2/A_1$. Therefore, this neuron functions as a detector of sound distance. Notably, the ITD and the travel time of the truck's honk differ, and $A_2/A_1$ is not inversely proportional to the square of ITD. Nonetheless, a clear mathematical relationship between them exists, as depicted in Supplementary Fig. 18. Hence, it is possible to determine the distance to the sound source based on our neurons' coincidence detection with an analog coincidence degree $A_2/A_1$ (refer to Supplementary Note 6 for further details).

In accordance with Fig. 5e, a hardware neural network was constructed (Fig. 5f). To emphasize the sound localization capability of our network, we neglected the section that emulates human ears to convert sound signals into electrical signals. Instead, we stimulated the output signals of sound sensing neurons with electrical signals that have different time intervals. Additionally, we substituted the modulation of synaptic weights on the spike amplitude of the sound signal with the modulation of voltage pulse amplitude using voltage divider circuits while retaining the diagonal synaptic weight matrix. The feasibility and efficacy of this approach, the modulation of synaptic weight on pulse spike amplitude, is depicted in Supplementary Fig. 19. Three dual-gate V-EGTs serve as the three post-neurons, with two green post-neurons working together on sound azimuth recognition and the yellow post-neuron functioning in sound distance recognition. The gates of all post-neurons function as dendrites.

Based on the aforementioned hardware neural network, whose equivalent circuit diagram is presented in Fig. 5g, we replicated the brain's recognition of sound azimuth (Fig. 5h–j) and distance (Fig. 5k), respectively. As an illustration of sound azimuth localization according to ITD, we displayed the intensity and sequence of sound signal spikes received by each dendrite of post-neurons 1 and 3, as well as the output signal when ITD = 2 s (Fig. 5h, i). The $I_{post1}/I_{post3}$ value corresponds to a unique ITD and can be utilized to determine the sound azimuth, thus simulating the brain's sound azimuth recognition function. By altering ITD, we further demonstrated the identification of all potential azimuths ($-90°$ – $90°$) (Fig. 5j).

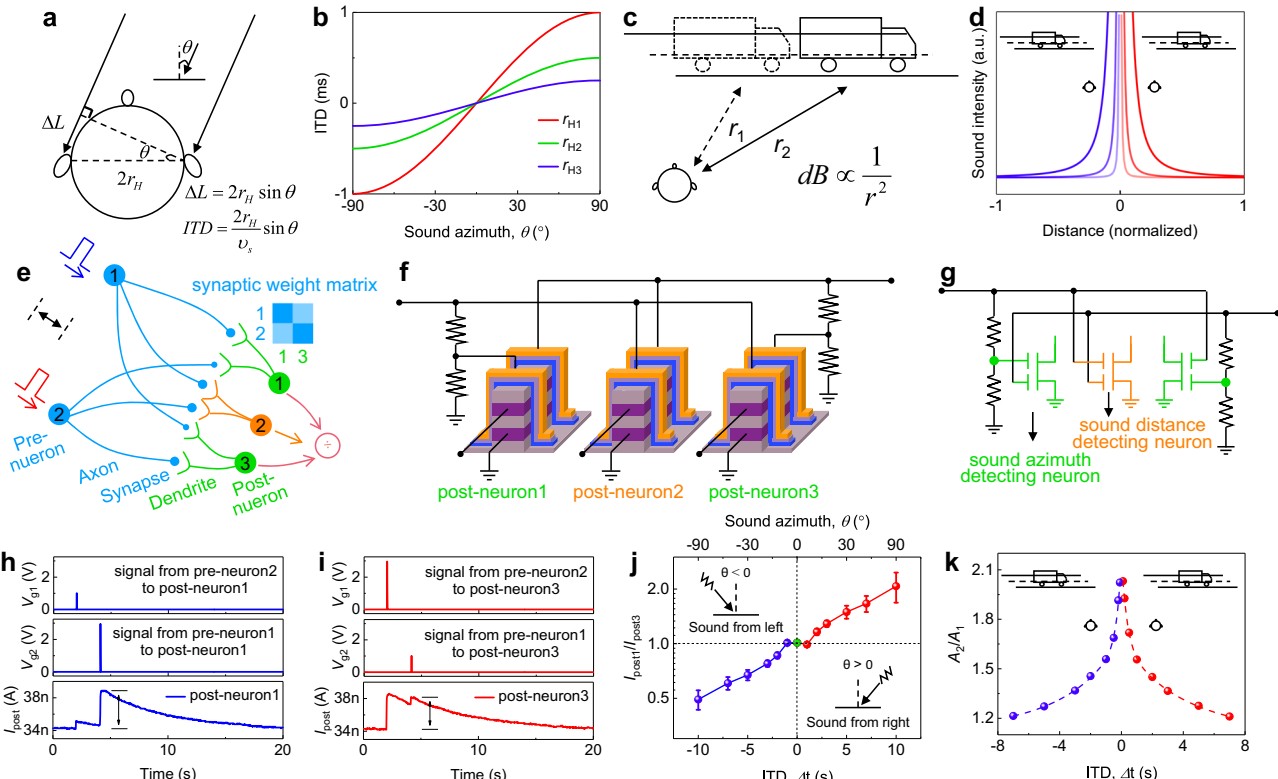

**Fig. 5 | Emulation of the brain's sound localization function based on dual-gate V-EGT. a** Principle of the brain recognizing sound azimuth, that is, based on the trigonometric relationship between interaural time difference (ITD) and sound azimuth θ, the brain can identify the sound azimuth through ITD. **b** Relationship between ITD and θ. **c** Principle of sound source distance computation by the brain. The farther the sound source is from the ear, the weaker the intensity of the sound heard by the ear. Specifically, the sound intensity is inversely proportional to the square of the distance. **d** Relationship between sound intensity and distance. **e** Schematic of the neural network for sound azimuth and distance identification. Blue spheres, blue lines, blue nodes, green or yellow lines, and green or yellow spheres represent (sound) sensing neurons, axons, synapses, dendrites, and post sound information processing neurons, respectively. The synaptic weight matrix between the two green post neurons and pre-neurons is diagonal. The yellow post-neuron has the same synaptic connection weights as two pre-neurons. **f** Hardware neural network for sound azimuth and distance recognition based on dual-gate V-EGTs. **g** Equivalent circuit diagram of (**f**). **h, i** An example of evaluating sound azimuth (ITD = 2 s). The intensity and sequence of sound signal spikes received by each dendrite of post neurons 1 and 3 are shown, as well as the outputs. **j** Recognition for all sound azimuths by our hardware neural network. **k** Emulation of the brain's sound source distance computation by our hardware neural network.

Aside from recognizing sound azimuth, we also demonstrated the capability of our hardware neural network to recognize sound source distance, based on the response of the yellow post-neuron to the left and right ears. By creating a symmetric dual-gate PPF of our dual-gate V-EGTs around ITD and integrating the inherent analog characteristic and nonlinear decay, our hardware neural network replicated the sound distance recognition function of the brain (Fig. 5k). Therefore, our constructed hardware neural network has the ability to simultaneously recognize both sound azimuth and distance.

The biological ITD typically falls within the range of 1 ms. However, due to the relatively long current decay time (-10 s) of our current V-EGT, our ITD for sound localization ranges from 0.1 to 10 s, which is significantly larger than its biological counterpart. Nonetheless, given that our current V-EGTs are capable of recognizing ITD within the range of 0.1–10 s, a future V-EGT with a decay time of 10 ms (refer to Supplementary Note 2 for further details) will be able to accurately identify ITD with smaller time differences (e.g., identify ITD with a 0.1 ms difference).

Sound localization in the human brain is not only characterized by high accuracy but also by fast processing speed, allowing for very short intervals between consecutive sound localizations. As for the current V-EGT prototypes with relatively long current decay times, there is a waiting period to process the next sound localization until the V-EGTs return to their initial states, implying a slow sound information processing. The strategies for optimizing sound localization accuracy (see Supplementary Note 2) are also contribute to improving the speed of

the sound localization system. Thus, reducing the decay time of V-EGTs not only benefits obtaining accurate ITDs but also facilitates the development of a fast sound localization system.

In conclusion, we have demonstrated an all-solid-state, vertical, compact and low-power EGT equipped with STM characteristics by engineering materials and devising a novel vertical structure. Our dual-gate V-EGT, benefiting from the multi-gates of the proposed vertical structure, functioned as a neurotransistor and successfully emulated the dendritic computing function of neurons, including dendrite integration and digital and analog computing for coincidence detection. By constructing a hardware neural network, we were able to largely replicate the sound localization function of the brain. Thinning the thickness of the electrolyte, optimizing the device's operating conditions, and constructing neural networks more similar to their biological counterpart could further improve the energy efficiency of our neurotransistors and enable more faithful emulations of the sound localization function of the brain. This work provides insight into replicating advanced cognitive functions of the brain with emerging neuromorphic systems that are of high density and low power consumption.

## Methods
### Construction of V-EGT
The 8-inch Si wafers were utilized to fabricate vertical sidewalls. Prior to fabrication, the substrate underwent a standard cleaning process to remove the native oxide layer on the surface of the Si wafer. Following

this, a layer of $SiO_2$ was formed on the Si wafer surface via thermal oxidation (1000 °C, 4 h).

## Preparation of the laminated structure

A four-layer structure of TiN + $SiO_2$ bilayer was prepared by sequential deposition of TiN (20 nm) and $SiO_2$ (30 nm) four times. TiN was deposited by PVD and served as the source and drain electrodes, while $SiO_2$ was deposited by plasma-enhanced chemical vapor deposition and used as a spacer between the source and drain electrodes.

## Construction of vertical sidewall

After patterning the Si wafer on which the laminated structure was grown (to create the sidewall and reserve positions for subsequent exposure of each electrode layer), the exposed part was dry etched with $BCl_3 + Cl_2$ gas using photoresist as a mask until it reached the substrate (the etching cavity temperature was 80 °C). Cleaning of the vertical sidewall structure was performed with a DSP cleaning agent to remove residual photoresist and organics generated during the etching process. The Si wafer was then dried with a nitrogen gun, making it suitable for the fabrication of V-EGT. After etching the vertical sidewalls, to facilitate subsequent electrical testing, it was necessary to expose the TiN located beneath the $SiO_2$ at different levels. Initially, the eight horizontal electrode pad positions in Fig. 1e had the same height, consisting of a $TiN/SiO_2/TiN/SiO_2/TiN/SiO_2/TiN/SiO_2$ stack. To expose the first TiN, we etched away the top $SiO_2$ at the five rightmost electrode pad positions in Fig. 1e, thereby exposing the first TiN electrode. Next, at the third electrode pad position from the left, we sequentially etched away the $SiO_2$, TiN, and $SiO_2$, exposing the second TiN electrode. This process was repeated, and we successively exposed the third (at the second electrode pad position from the left) and the fourth (at the first electrode pad position from the left) TiN electrodes. With this procedure, all the TiN electrodes covered by the $SiO_2$ were exposed, enabling convenient measurement of the electrical characteristics of V-EGT devices.

## Channel/electrolyte/gate stack deposition

Following the patterning of the Si wafer with vertical sidewall structure, α-$Nb_2O_5$ (~20 nm), $Li_xSiO_2$ (~44 nm), and Ti/Au were sequentially deposited. Magnetron sputtering was employed to deposit α-$Nb_2O_5$ and $Li_xSiO_2$, which served as the channel and electrolyte of V-EGT, respectively, while electron beam evaporation (EBE) was used to deposit Ti/Au, which acted as the gate. The V-EGT was released through a lift-off process. It should be noted that the above thicknesses for the three materials are the thicknesses of each material when deposited on a flat surface. As the thin film grown on the sidewall by magnetron sputtering or EBE is thinner than the thin film grown on the flat surface, the actual thickness of each functional layer of V-EGT should be calibrated according to the cross-sectional TEM image of the corresponding device.

## Electrical characterization

The electrical characteristics of V-EGT were obtained at room temperature in atmospheric conditions using a semiconductor parameter analyzer B1500A.

## Data availability

All data needed to evaluate the conclusions in the paper are present in the paper and/or the Supplementary Materials. Additional data related to this paper may be requested from the authors.

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

## Acknowledgements

This work was supported by the National Key R&D Program of China under Grant No. 2018YFA0701500, the National Natural Science Foundation of China under Grant Nos. 62374181, 61825404, 61732020, 61851402 and 61974081, the Strategic Priority Research Program of the Chinese Academy of Sciences under Grant XDB44000000, the China Postdoctoral Science Foundation under Grant 2023M731885. The authors thank D. He (Lanzhou University) for the electrical characterization.

## Author contributions

D.S. and Q. Liu conceived the idea and designed the experiments and simulations. H.X, D.S. and Q. Luo fabricated the samples, while H.X., Z.Y, Y.L. and S.W. performed the electrical measurements. J.A. and W.Z. carried out the simulations. X.X., C.D., H.J., L.P., X.Z., M.W. and J.T. participated in the discussion of results. D.S., Q. Liu, and M.L. supervised the entire work. D.S, H.X. and Z.W. wrote the manuscript. All authors commented on the manuscript.

## Competing interests

The authors declare no competing interests.

## Additional information

**Supplementary information** The online version contains
supplementary material available at

Dashan Shang, Jianshi Tang or Qi Liu.

**Peer review information** *Nature Communications* thanks Gunuk Wang,
and the other, anonymous, reviewers for their contribution to the peer
review of this work. A peer review file is available.

