## [Peer Review File · Nature Communications]

This paper delineates the creation of a compact, low-power neurotransistor using a V-EGT with STM characteristics. The author conceptually demonstrated dendrite integration and digital/analog dendrite computing for coincidence detection. It is shown that the V-EGT permits easy modification of the device and switching parameters, thereby enhancing performance, a feat not achievable with the planar type of neurotransistor. The switching parameters have been thoroughly investigated under various conditions. Furthermore, the vertically stacked EGT presented exhibits the lowest read power among all inorganic-EGT devices. The paper is well-written and understandable. However, some uncertainties in the paper necessitate additional explanations before it can be considered ready for publication.

(1) It seems that the integration density of V-EGT structure is dependent on the gap distance between the patterned gate electrodes. If the gate electrodes are close to each other to a certain point, the channel conductance might be strongly disturbed and its control capability is affected. I think that using electrolytes as a dielectric layer would be affected more. I suggest that the author should discuss it with potential issues in the main text.

(2) The author highlighted that in the V-EGT structure, it is possible to make multiple channels stacked on the substrate. However, in Figs. 2-5, there is no utilization of vertically stacked transistor structure (multiple channel layers) as well as the analysis of the sufficient electrical characteristics. Only, double gate structure-based electrical characteristics and utilization are mainly shown. Therefore, I believe that another essential of V-EGT with multiple channel layers is ignored. Additional experiments and explanations are required. And, discussion on what neuromorphic electronic applications are possible is needed.

(3) In order to replicate the essential of signal propagation through the dendrites in a biological neural network, it is important to implement distinct and independent synaptic clefts with individual synaptic weights. However, this V-EGT structure shares the same channel where identical gate pulses are applied to distinct gate electrodes. I believe that this device structure is operable with various gate pulses and timings (making different and individual synaptic weights) that make various channel conductance. It is necessary to show the switching characteristics with various operating schemes for the dendrite computing capability.

(4) This device utilized the electrolyte. The temperature condition would have a significant effect. Display and discuss the temperature-dependent electrical characteristics (decay degree and conductance change, etc.).

(5) I am a bit confused about understanding Fig. 3b. Would you explain it more clearly?

(6) For the power consumption, the author only considered the read power. Although the gate-source path is capacitive and generates very low leakage current, V_D is still applied when programming, which could generate the programming power. The author should consider this. What is the power consumption when the programming? And compare it with other literatures.

(7) In Fig. 3d, the author said that the gate leakage current is lower than drain current, but in log scale, there is no significant distinguishable difference.

(8) What is gate pulse condition in Fig. 3f?

(9) In Fig. 4g. I couldn't find any values regarding $\Delta t = 7s$.

(10) The author showed a sound localization function using dual-gate PPF. Here, it is doubtful whether there will be any problem in processing the next information in a state where relaxation has not occurred completely.

Reviewer #2 (Remarks to the Author):

The manuscript entitled “A low-power vertical dual-gate neurotransistor with short-term memory for high energy-efficient neuromorphic computing” by Xu et al. deals with a high-performing neuromorphic device, whose challenge aims to mimic part of the brain functions.

The manuscript is satisfactorily written and well-organized. Although my evaluation is positive, different issues must be addressed before the publication of this manuscript.

- 1) Albeit it is clear the advantages of the V-EGTs compared to the planar one, the fabrication of this device consists of the deposition of 8 layers versus the 4 ones of the planar. Please define better the materials consumption in terms of the scalability of the process.
- 2) One fundamental aspect of this V-EGT is the reduced size of the footprint, however, it is clear that the overall dimension of the V-EGT is similar or even larger than the planar one. Can you add information on the possibility of miniaturizing it?
- 3) There is missing information in the experimental section: since there are 4 electrodes embedded into SiO₂, how are the connections established experimentally? Do they impact the scalability of this type of device? Figure 1c and Figure 1e only show the position of these connections.
- 4) Figure S7 is not mentioned in the main text. Furthermore, a short paragraph is required to describe properly these schemes.
- 5) Figure 4d is not mentioned in the main text. Please add a corresponding sentence.
- 6) The following sentence “In this setup, the gate (drain) of a dual-gate V-EGT is considered as the dendrite (axon) of a neuron, and our device can mimic signal propagation pathways in the brain” should be rephrased, because it is not clear the role of the gate/drain. Gate and drain have two distinct roles in the device operation.

Reviewer #3 (Remarks to the Author):

The manuscript by Xu et al. describes a low-power synaptic transistor which can be used for dendritic computing. They address challenges in EGT fabrication by using a vertical channel which reduces the device footprint and facilitates fabrication dense arrays without added complexity/lithography steps. The devices are well characterized and show impressive low energy operation. The manuscript also includes a demonstration of devices used for coincidence detection which clearly demonstrates the utility of the presented devices.

Overall, the work presents a significant advance in the development of synaptic transistors and warrants publication in Nature Communications after minor revisions. I have added a few technical comments that I would like to see addressed before the paper can be accepted for publication.

On line 42: the size of the human brain is compared to a football, but this size means different things depending where in the world the reader is from. I suggest also including a number for the rough volume of the brain.

On line 49: It would be helpful to clarify this motivation, as digital hardware does not necessarily “suffer” from the slowdown Moore’s Law, instead digital hardware is not improving at an exponential pace anymore, limiting the ability to make increasingly complex ANNs without increasing compute times.

On line 82: the channel length of 30 nm is not a representative number for the device footprint since it is in the vertical direction. Is there a fundamental limitation to the process that can be quoted here? Is the footprint limited by the lithography? Or the directionality of the etching step? The resistivity of the TiN source/drain contacts or channel?

Is the long-term memory (shown in Fig. 2c,h,f) a permanent change in the channel conductance? Or is the transient very slow and will reach the original baseline conductance given enough time? If it is permanent, what is the difference in switching mechanism between the short-term and long-term memory processes?

The devices still operate quite slowly compared to other artificial synapses cited in the paper/SI. Is there a path towards pushing these materials to write time below 1 μ s? The projections in S14 are nice, but it seems it would be extremely challenging to scale electrolyte thickness down to the level necessary for μ s operation.

Since the capacitance of the electrolyte depends on the area of overlap, would the switching speed or the device scale with the channel width? Could the gate contact geometry be optimized to improve switching speeds? Or are there other strategies to increase speeds to $< 1 \mu$ s?

Is there crosstalk between devices in the bilayer structure? Or is the read voltage so low that it does not disturb the resistance state of the adjacent channel?

We sincerely appreciate the editor's time in handling our manuscript and are delighted that reviewers are positive about our work. We are grateful to the reviewers for their valuable comments and have addressed their comments point-by-point as listed below. The changed parts in the manuscript have been highlighted in yellow in the revised manuscript.

Comment from the Reviewer #1

This paper delineates the creation of a compact, low-power neurotransistor using a V-EGT with STM characteristics. The author conceptually demonstrated dendrite integration and digital/analog dendrite computing for coincidence detection. It is shown that the V-EGT permits easy modification of the device and switching parameters, thereby enhancing performance, a feat not achievable with the planar type of neurotransistor. The switching parameters have been thoroughly investigated under various conditions. Furthermore, the vertically stacked EGT presented exhibits the lowest read power among all inorganic-EGT devices. The paper is well-written and understandable. However, some uncertainties in the paper necessitate additional explanations before it can be considered ready for publication.

Reply:

We sincerely appreciate the reviewer's positive evaluation for our work. These insightful comments and suggestions have significantly improved the quality of our paper. We have provided point-by-point response as follows.

Comment 1:

It seems that the integration density of V-EGT structure is dependent on the gap distance between the patterned gate electrodes. If the gate electrodes are close to each other to a certain point, the channel conductance might be strongly disturbed and its control capability is affected. I think that using electrolytes as a dielectric layer would be affected more. I suggest that the author should discuss it with potential issues in the main text.

Reply to Comment 1:

Thanks for your comment. Indeed, as correctly pointed out by the reviewer, the integration density of V-EGT is strongly influenced by the distance between adjacent gate electrodes. In the revised manuscript, we have included a discussion on the area of V-EGT devices, particularly in the case of array configurations. The following discussion has been incorporated into the revised manuscript:

In the main text, the added content is as follows (See page 13 in the main text, line 12-14):

“(see Supplementary Note 3 for more discussion on the calculation of V-EGT device footprint in array configuration)”

In the Supplementary Materials, the detailed discussion is as follows (see page 31-33 in the Supplementary Materials):

“For V-EGT, the effective device area is considered to be the projection area of the core region on the horizontal plane. Therefore, the area of V-EGT is determined by the electrode width of the source/drain and the gate width. It can be calculated by the product of the source/drain electrode width and the gate width. Accurately, it should be twice the source/drain electrode width multiplied by twice the gate width, considering the spacing of one feature size between adjacent source/drain electrodes and adjacent gates (Supplementary Fig. 20).

The device area of V-EGT is limited by the lithographic precision which determines both the source/drain electrode width and the gate width. In our experiments, the typical source/drain electrode width is 10 μm , and the minimum gate width is 2 μm . Therefore, the smallest achieved area of our V-EGT is $10 \mu\text{m} \times 2 \mu\text{m} = 20 \mu\text{m}^2$. Within the capability of ultraviolet lithography, the source/drain electrode width can be reduced to a minimum of 1 μm , and the gate width can also be reduced to a minimum of 1 μm . Hence, the minimum device area of our V-EGT can reach $1 \mu\text{m} \times 1 \mu\text{m} = 1 \mu\text{m}^2$. Of course, when electron beam lithography is used, the device area of V-EGT can be further reduced.

Supplementary Fig. S20. Calculation of V-EGT device footprint in array configuration. a) 3D schematics of V-EGT. **b)** Top-down view of **a**.

From a manufacturing process perspective, the etching process still have some influence on the device area of V-EGT, as the width of the source/drain electrode, obtained through etching, is easier to achieve in a larger dimension (e.g., $10\ \mu\text{m}$) compared to a smaller dimension (e.g., $100\ \text{nm}$). Furthermore, the resistance of TiN source/drain electrodes is also a limiting factor in the device area of V-EGT due to the finite resistivity of TiN. The TiN source/drain electrodes must have a sufficient width to ensure proper device operation. This prerequisite restricts the source/drain electrode width from being arbitrarily small.

In V-EGTs, especially in their array configuration form, mutual interference between adjacent gate electrode stacks has to be considered. There are two forms of potential mutual interference between adjacent gate electrode stacks. The first form occurs when different gate electrodes share the same channel and electrolyte layer (Fig. S21b). In this situation, interference between adjacent gate electrode stacks is inevitable, with the severity of the interference increasing as the distance between them decreases. Since electrolyte ions can freely move throughout the electrolyte layer, they can capacitively couple to the same channel, regardless of the distance between different gate electrodes. This capacitive coupling between the gate electrode and the channel is

strongly influenced by the distance between them. Consequently, when two gate electrodes are in close proximity, the gate control capability of one electrode over the channel will be significantly affected by the presence of the other electrode. Fortunately, this form of mutual interference is prevented in our case, as different channel/electrolyte/gate stacks are deposited separately (Fig. S21a).

Supplementary Fig. S21. Two structures of dual-gate V-EGT. **a)** Dual-gate V-EGT with independent electrolytes and channels. **b)** Dual-gate V-EGT with common electrolytes and channels.

The second type of mutual interference between adjacent gate electrode stacks can be attributed to a parasitic capacitance between nearby conductors. This parasitic capacitance is inherent in almost all 3D-integrated circuits with metal interconnects, and the capacitive coupling between metals or conductors becomes stronger as the distance between them decreases. This effect poses a challenge for miniaturization and high-density array integration. Likewise, when the distance between adjacent gate electrodes becomes sufficiently small, the parasitic capacitance effect of the V-EGT will significantly impact the gate control capability.”

Comment 2:

The author highlighted that in the V-EGT structure, it is possible to make multiple channels stacked on the substrate. However, in Figs. 2-5, there is no utilization of vertically stacked transistor structure (multiple channel layers) as well as the analysis of the sufficient electrical characteristics. Only, double gate structure-based electrical characteristics and utilization are mainly shown. Therefore, I believe that another essential of V-EGT with multiple channel layers is ignored. Additional experiments and

explanations are required. And, discussion on what neuromorphic electronic applications are possible is needed.

Reply to Comment 2:

We thank the reviewer for the constructive suggestion and appreciate their insight regarding the untapped potential of the multi-channel characteristics of V-EGT. We agree that, in addition to the multi-gate characteristics, the multi-channel length feature is another important aspect of V-EGT. Similar to the neuromorphic computing application showcased in this manuscript, which is driven by the multi-gate characteristics, we believe that the multi-channel characteristics of V-EGT will also excel in other neuromorphic computing applications. We are actively working our next study in this direction and have already obtained experimental data.

This manuscript primarily focuses on the short-term memory and multi-gate characteristics of V-EGT, demonstrated in neuromorphic computing applications such as dendritic computing in neurons and sound localization in the brain (**Figs. 2, 4, and 5**). We also provide a basic characterization of the device structure and low-power features of V-EGT (**Figs. 1 and 3** in the main text). Furthermore, there are extensive data around the short-term memory and multi-gate characteristics of V-EGT in Supplementary Materials. Nevertheless, some data related to the multi-channel characteristics of V-EGT can be found in the Supplementary Materials Fig. S8, which complements the aforementioned characteristics and applications.

Our upcoming study primarily explores the multi-channel characteristics of V-EGT, which is distinct from the multi-gate characteristics and plays different roles in neuromorphic computing. In summary, the multi-channel characteristics of V-EGT and their neuromorphic computing applications can form an independent investigation, which is currently underway. We emphasize that this study will not impact the elucidation of the issues in our current work, specifically the application of multi-gate STM V-EGT in dendritic computing and sound localization.

Nevertheless, we apologize for being unable to present the content related to our upcoming work at this stage. We hope for the reviewer's understanding and look

forward to sharing our findings in the future.

Supplementary Fig. S8. Variable channel length characteristics of V-EGT. a) - c) Transfer characteristic curves of V-EGT with different source-drain electrode selections at channel lengths of 30, 80, and 130 nm, respectively. The scale bar is 50 nm.” (See page 11 in the Supplementary Materials, line 1-4)

Comment 3:

In order to replicate the essential of signal propagation through the dendrites in a biological neural network, it is important to implement distinct and independent synaptic clefts with individual synaptic weights. However, this V-EGT structure shares the same channel where identical gate pulses are applied to distinct gate electrodes. I believe that this device structure is operable with various gate pulses and timings (making different and individual synaptic weights) that make various channel conductance. It is necessary to show the switching characteristics with various operating schemes for the dendrite computing capability.

Reply to Comment 3:

We sincerely appreciate the reviewer's insightful comment. We concur with the reviewer that, for a more realistic simulation of dendritic computing in biological neurons, each dendrite of the dual-gate V-EGT neuron should possess distinct synaptic weights. Specifically, at the device level, each gate of the dual-gate V-EGT should

exhibit different gate control capabilities for their respective channels. Due to the identical material dimensions of the channel/electrolyte/gate stacks in the dual-gate V-EGT used in this work, we could not mimic the characteristics of different dendrites with varying synaptic weights, as observed in biological neurons. However, this does not imply that the dual-gate V-EGT is incapable of achieving this attribute. By depositing separate channel/electrolyte/gate stacks with distinct material dimensions, such as channel thickness, channel width, electrolyte thickness, and electrolyte width, the dual-gate V-EGT devices can indeed demonstrate characteristics similar to those of biological neurons. This indicates that the dual-gate V-EGT has the potential to attain characteristics akin to biological neurons by adjusting the device fabrication process.

Furthermore, the dual-gate V-EGT exhibits varying current responses when subjected to different gate pulse amplitudes and widths, despite possessing the same synaptic weight. Examples of this can be found in **Figs. 4d** and **e**, where extreme cases are displayed, i.e., one gate is subjected to voltage while the other gate remains at zero voltage. In addition, different gate voltage intensities for the same dual-gate V-EGT are achieved indirectly, as illustrated in **Figs. 5h** and **i**, wherein a voltage divider circuit is employed to modify the applied gate voltage intensities. Different pulse intensities can also be directly applied to distinct gates of the same dual-gate V-EGT. We have incorporated the relevant content into the revised manuscript's main text and Supplementary Materials, which we have provided below for the reviewer's evaluation.

In the main text, the added content is as follows (See page 15 in the main text, line 1-8 from the bottom):

“Note that, the dual-gate V-EGT devices are not exact the same to the biological neurons. The different gate stacks of the devices have the same synaptic weight, while different dendrites of biological neurons inherently have different synaptic weights. This characteristic of biological neurons can be achieved indirectly by applying different pulses to each gate of the dual-gate V-EGT. (**Supplementary Fig. S17**). One possible way to achieve the synaptic weights with inherent difference is separately depositing the different channel/electrolyte/gate stacks of the dual-gate V-EGT with

different material dimensions.”

In the Supplementary Materials, the added content is as follows:

“

Supplementary Fig. S17. Current-time curves of dual-gate V-EGT in response to different gate pulse a) amplitudes and b) widths.” (See page 20 in the Supplementary Materials,

line 1-3)

Comment 4:

This device utilized the electrolyte. The temperature condition would have a significant effect. Display and discuss the temperature-dependent electrical characteristics (decay degree and conductance change, etc.).

Reply to Comment 4:

We appreciate the reviewer’s comments and agree that temperature has a significant impact on device characteristics, particularly when the device contains electrolyte materials. It is important to note that unlike typical proton and oxygen ion conductors where temperature mainly affects their ionic conductivity with minimal impact on other material properties, lithium-ion conductors can undergo reactions with oxygen and moisture in the air due to their high reactivity. This can lead to irreversible changes in devices.

As mentioned in the main text, the industry is actively exploring methods to address the inherent material instability and incompatibility with CMOS processes of

lithium-based electrolyte materials. These methods include using TiN barrier layers to prevent contamination of other CMOS materials by lithium-based electrolytes, and passivation or encapsulation of lithium-based electrolyte-gated transistors (Li-EGTs). However, these processing steps have not been implemented in the current work due to the complexity of the dual-gate V-EGT device preparation process.

We have conducted measurements on the temperature characteristics of the EGT with planar structure. **Fig. R1** shows the retention characteristics of the EGT devices at different temperatures, measured under gate-source short-circuit conditions. As temperature increases, decay processes are accelerated due to the enhancement of ion migration. The retention characteristics should exhibit a consistent trend with temperature between the EGTs and the V-EGTs.

For future developments, we suggest that materials with enhanced temperature robustness are required. High-temperature tolerant EGT materials and devices have been a subject of intense research recently (Melianas *et al.*, *Adv. Funct. Mater.*, 2021). Additionally, regarding the overall computational system, designing dependable peripheral circuits for temperature compensation might be another potential solution. Since this work primarily focuses on the dual-gate V-EGT device preparation and application demonstration in dendritic computing and sound localization, temperature-related investigations are beyond the scope of the current study.

Figure R1. Retention characteristics of the planar EGT at different temperatures. The increasing and then decaying in channel conductance at 25 °C can be attributed to slow interface kinetics and ion redistribution (refer to the Review by Jesús del Alamo, Ju Li, and Bilge Yildiz

for further details; Huang et al., Adv. Mater., 2023).

Comment 5:

I am a bit confused about understanding Fig. 3b. Would you explain it more clearly?

Reply to Comment 5:

Thank you for the reviewer’s comment. We are happy to provide a clearer explanation of **Fig. 3b** of the original manuscript. Under the same channel current conditions (with only drain voltage and no gate voltage) and constant channel width and thickness, the

current formula is $I_D = \frac{S_{channel} V_D}{\rho_{channel} L_{channel}}$ (derived from the combination of formulas

$$I_D = \frac{V_D}{R_{channel}} \text{ and } R_{channel} = \rho_{channel} \frac{L_{channel}}{S_{channel}}),$$

where I_D is the channel current, V_D the read voltage, $R_{channel}$ the channel resistance, $\rho_{channel}$ the channel resistivity, $L_{channel}$ the channel length, and $S_{channel}$ the cross-sectional area of the channel along the current direction. That means the shorter the channel length, the smaller the read voltage is required. Under the same channel current conditions, devices with shorter channels allow for a significant reduction in the required read voltage, thereby significantly reducing the read power and energy consumption of the device. This aligns with our pursuit of low-power operation for the devices. The original **Fig. 3b** and its related content are provided below for the reviewer's reference.

Figure 3 b. Relationship between V_D and $L_{channel}$ with fixed channel thickness, width, and channel current. Situations with different channel currents are shown.

In the main text, the related content is as follows:

“With the channel thickness, width, and current remaining constant, V_D is calculated to decrease linearly with the decrease in channel length (Fig. 3b). Therefore, V-EGTs can achieve a substantial reduction in their read voltage, and subsequently, their read power consumption.” (See page 10 in the main text, line 1-2 from the bottom for the figure caption of Fig. 3b and page 11 in the main text, line 5-8 from the bottom for the corresponding content in the main text)

Comment 6:

For the power consumption, the author only considered the read power. Although the gate-source path is capacitive and generates very low leakage current, V_D is still applied when programming, which could generate the programming power. The author should consider this. What is the power consumption when the programming? And compare it with other literatures.

Reply to Comment 6:

Thank you for the insightful comment. We agree with the reviewer's perspective that, if read voltages are still present during programming, the energy consumption generated by the read operation should be added to the programming energy consumption. However, we are unable to incorporate such read energy into our write energy consumption calculation because the write and read operations of EGT can be effectively separated in time, similar to the approach adopted by Fuller *et al.* Consequently, the read operation will have no impact on the write energy consumption of EGT. Meanwhile, to ensure fair comparison with existing literatures, we have not accounted for the influence of V_d on the programming energy consumption of EGT.

We employ the formula $E_w = I_w \times V_w \times t_w$ to calculate the programming energy consumption of V-EGT, where E_w , I_w , V_w , and t_w represent the programming energy, programming current, programming voltage, and programming time of the write operation, respectively. As an example, we estimate the programming energy

consumption of V-EGT based on Supplementary Fig. S13c of the original Supplementary Materials (see Fig. R2 here). The key challenge is to measure accurately the programming current, which is often very small. On the other hand, it usually requires the construction of dedicated and sophisticated test circuits to measure the programming current.

Figure R2. Channel current-time response curve at read voltage of 0.1 mV.

We have made the following modifications in the revised manuscript:

“Apart from the read energy, write energy is also an important aspect of EGT's energy profile, although it can be neglected due to the minimal gate leakage current. The key challenge in write energy estimation is to measure accurately the write current. By assuming an average gate leakage current of 50 pA (the maximum value observed in Fig. 3d), based on Supplementary Fig. S13c, we have estimated the write energy of V-EGT (275 fJ = 50 pA \times 5.5 V \times 1 ms) and compared it with other literatures (Supplementary Table 3). (See page 13 in the main text, line 3-9)

We have made the following modifications in the Supplementary Materials:

Year	Reference	I_w	V_w (V)	t_w	Write energy (J)
2023	this work	50 pA(assumed)	5.5	1 ms	275 fJ(assumed)
2022	Ref. 3	NA	-0.7	0.1 s	NA
2021	Ref. 6	NA	6.5	10 μ s	NA
2020	Ref. 5	-79.1 nA 15.9 nA	-3 V 2 V	50 ms 50 ms	11.9 nJ 1.6 nJ
2019	Ref. 4	NA	2 V	10 ms	NA
2019	Ref. 2	NA	0.6	500 ms	NA
2016	Ref. 7	NA	-1 V	50 ms	NA
2021	Ref. 8	NA	1 V	100 ns	NA
2019	Ref. 9	NA	2 V	200 ns	NA
2017	Ref. 10	<1 μ A	1 mV	NA	~10 pJ
2020	Ref. 11	0.5 μ A	0.25 V	5 ms	625 pJ
2020	Ref. 12	NA	4 V	NA	~1 pJ
2019	Ref. 13	9 nA	0.3 V	10 ms	~30 pJ
2018	Ref. 14	100 nA(projected)	NA	10 ns(projected)	1 fJ(projected)
2016	Ref. 15	NA	NA	1 μ s(projected)	<1 aJ(projected)
2021	Ref. 16	NA	1 V	4 μ s	1.6 nJ
2018	Ref. 17	NA	NA	NA	500 fJ
2018	Ref. 18	NA	2.5 V	1 ms	NA
2017	Ref. 19	NA	2.5 V	1 ms	NA

Supplementary Table T3. Comparison among various EGTs in terms of write energy³⁻²⁰.

(See page 27 in the Supplementary Materials, line 1-2)

Comment 7:

In Fig. 3d, the author said that the gate leakage current is lower than drain current, but in log scale, there is no significant distinguishable difference.

Reply to Comment 7:

We agree with the reviewer's comment. The statement that the channel current of the V-EGT is "significantly larger" than the gate leakage current, is imprecise. As the reviewer correctly pointed out, there may not be a significant difference between the channel current and gate leakage current in a logarithmic scale. To address this, we have

removed the term "significantly" in the revised manuscript. The revised text is provided below for the reviewer's reference.

“Moreover, the channel current was found to be obviously larger than the gate leakage at $V_D = 0.1$ mV (~ 150 pA Vs. ~ 50 pA), thereby validating the effectiveness of the 0.1 mV read voltage (Fig. 3d).” (See page 11 in the main text, line 1-4 from the bottom)

Comment 8:

What is gate pulse condition in Fig. 3f?

Reply to Comment 8:

Thanks for the reviewer's comment. The gate pulse conditions for Fig. 3f were $V_W = 4$ V and $t_W = 10$ ms. We have included them in the revised figure caption of Fig. 3f. Figure 3f and its modified figure caption are provided below for the reviewer's reference.

“

Figure 3 f. Channel current-time response of the V-EGT at different read voltages. The inset zooms in on the curve corresponding to 1 mV read voltage. The gate pulse condition is $V_W = 4$ V, $t_W = 10$ ms.” (See page 11 in the main text, line 5)

Comment 9:

In Fig. 4g. I couldn't find any values regarding delta t = 7s.

Reply to Comment 9:

Thanks for the reviewer's comment. We provide the corresponding data for $\Delta t = 7$ s in Fig. 4g (referred to as Fig. R3 here). In each color-coded subfigure within the bottom panel of Fig. 4g, the bars with the smallest magnitudes represent the data corresponding

to $\Delta t = 7\text{s}$. Here, we have provided a visual representation to clearly indicate the location of the data corresponding to $\Delta t = 7\text{s}$ (see Fig. R3).

Figure R3. Digital and analog computing for coincidence detection of neurons based on dual-gate V-EGTs, in which the data corresponding to $\Delta t = 7\text{s}$ has been highlighted.

Comment 10:

The author showed a sound localization function using dual-gate PPF. Here, it is doubtful whether there will be any problem in processing the next information in a state where relaxation has not occurred completely

Reply to Comment 10:

We appreciate the insightful comment from the reviewer. It is true that the impact of EGT's decay time on subsequent sound localization is very important. For practical biomimetic sound localization, fast sound information processing is equally important to high sound localization accuracy. In our demonstration, achieving fast sound information processing requires EGT devices with short current decay times, enabling rapid restoration to the initial state after completing a sound localization task and preparing for the next one. Therefore, the reviewer's comment will help us with a more comprehensive consideration when utilizing our devices to simulate the sound localization functionality of the brain. We have included the discussion on the influence of current decay time on sound information processing speed in the revised manuscript, and the relevant content is provided below for the reviewer's convenience.

"Sound localization in the human brain is not only characterized by high accuracy but also by fast processing speed, allowing for very short intervals between consecutive

sound localizations. As for the current V-EGT prototypes with relatively long current decay times, there is a waiting period to process the next sound localization until the V-EGTs return to their initial states, implying a slow sound information processing. The strategies for optimizing sound localization accuracy (see Supplementary Note 2) are also contribute to improving the speed of the sound localization system. Thus, reducing the decay time of V-EGTs not only benefits obtaining accurate ITDs but also facilitates the development of a fast sound localization system." (See page 21 in the main text, line 1-9)

Comment from the Reviewer #2

The manuscript entitled "A low-power vertical dual-gate neurotransistor with short-term memory for high energy-efficient neuromorphic computing" by Xu et al. deals with a high-performing neuromorphic device, whose challenge aims to mimic part of the brain functions. The manuscript is satisfactorily written and well-organized. Although my evaluation is positive, different issues must be addressed before the publication of this manuscript.

Reply:

We sincerely appreciate the reviewer's positive evaluation for our work. These insightful comments and suggestions have significantly improved the quality of our paper. We have provided point-by-point response as follows.

Comment 1:

Albeit it is clear the advantages of the V-EGTs compared to the planar one, the fabrication of this device consists of the deposition of 8 layers versus the 4 ones of the planar. Please define better the materials consumption in terms of the scalability of the process.

Reply to Comment 1:

Thank you for the valuable comment. As the reviewer pointed out, it is true that V-EGTs require more material consumption compared to planar EGTs. Planar EGTs typically

consist of four layers, including the source/drain electrode, channel, electrolyte, and gate. On the other hand, V-EGTs involve seven layers, which are the electrode/insulator/electrode/insulator stack (specifically TiN/SiO₂/TiN/SiO₂), channel, electrolyte, and gate. Therefore, our V-EGT implementation requires three additional layers compared to planar EGTs. We have made the corresponding modifications in the caption of **Supplementary Fig. 1** in revised Supplementary Materials and provided the relevant content below for the reviewer's reference.

“V-EGTs indeed require more material consumption compared to planar EGTs. Planar EGTs consist of four layers, that is, source/drain electrode, channel, electrolyte, and gate, while the V-EGTs involve seven layers, that is, electrode/insulator/electrode/insulator stack (specifically TiN/SiO₂/TiN/SiO₂), channel, electrolyte, and gate. In the V-EGT design, it is worth noting that the topmost layer of SiO₂ in the TiN/SiO₂/TiN/SiO₂ stack can be omitted (similar to the approach adopted by Duan *et al.* in vertical IGZO TFTs¹), reducing the materials consumption to six layers. Although V-EGTs require more materials consumption, they can achieve ultrashort channel lengths in a simple and cost-effective manner, whereas planar EGTs would require expensive lithography equipment (such as electron beam lithography) to achieve sub-100-nanometer channel lengths.” (See page 2 in the Supplementary Materials, line 1-9 from the bottom and page 3 in the Supplementary Materials, line 1-2)

Comment 2:

One fundamental aspect of this V-EGT is the reduced size of the footprint, however, it is clear that the overall dimension of the V-EGT is similar or even larger than the planar one. Can you add information on the possibility of miniaturizing it?

Reply to Comment 2:

We appreciate the reviewer's comment. The perception that our V-EGTs have comparable or even larger device areas than planar EGTs may stem from the optical micrograph image of a four-layer dual-gate V-EGT, as shown in **Fig. R4a** (**Fig. 1e** in

the original manuscript). It should be noted that Fig. R4a depicts a V-EGT with two gates. In a single V-EGT, only one gate is required, which means that the area of an individual V-EGT can be reduced by at least half compared to the image in Fig. R4a. Additionally, considering the source/drain electrodes of the V-EGT, that is, the 8 horizontal electrodes in Fig. R4a. In a single V-EGT, only two of the eight electrode positions shown in Fig. R4a are needed—one for the exposed source electrode position and the other for the exposed drain electrode position. In particular, when the exposed positions of the two horizontal electrodes on the left and right sides of the gate are used as the source and drain positions, respectively, the area of a single V-EGT can be reduced to one-eighth of the area shown in Fig. R4a. Under such circumstances, a single V-EGT and a single planar EGT have the same arrangement of source, drain, and gate pads, with the gate located in the middle and the source and drain electrodes placed on the left and right sides (Fig. R4b, c). Given that the V-EGT and planar EGT have the same arrangement of source, drain, and gate pads, the V-EGT would have the same device footprint as the planar EGT when the pad area is equivalent. Therefore, the V-EGT does not have a larger device area than the planar EGT.

Figure R4. Optical micrograph images or schematic diagrams of various EGT structures.
a. Four-layer dual-gate V-EGT. **b.** Single-layer single-gate V-EGT. **c.** Planar EGT.

We further discuss how to reduce the device footprint of V-EGT. First, we need to clarify how to define the device footprint of V-EGT, especially in the case of an array configuration. We have added corresponding discussions in the revised manuscript's main text and Supplementary Materials, which are provided below for the reviewer's reference.

In the main text, we have added the following content:

“(see **Supplementary Note 3** for more discussion on the calculation of V-EGT device footprint in array configuration)” (See page 13 in the main text, line 12-14)

In the Supplementary Materials Note 3, we have added the following content:

“For V-EGT, the effective device area is considered to be the projection area of the core region on the horizontal plane. Therefore, the area of V-EGT is determined by the electrode width of the source/drain and the gate width. It can be calculated by the product of the source/drain electrode width and the gate width. Accurately, it should be twice the source/drain electrode width multiplied by twice the gate width, considering the spacing of one feature size between adjacent source/drain electrodes and adjacent gates (**Supplementary Fig. 20**).

The device area of V-EGT is limited by the lithographic precision which determines both the source/drain electrode width and the gate width. In our experiments, the typical source/drain electrode width is $10\ \mu\text{m}$, and the minimum gate width is $2\ \mu\text{m}$. Therefore, the smallest achieved area of our V-EGT is $10\ \mu\text{m} \times 2\ \mu\text{m} = 20\ \mu\text{m}^2$. Within the capability of ultraviolet lithography, the source/drain electrode width can be reduced to a minimum of $1\ \mu\text{m}$, and the gate width can also be reduced to a minimum of $1\ \mu\text{m}$. Hence, the minimum device area of our V-EGT can reach $1\ \mu\text{m} \times 1\ \mu\text{m} = 1\ \mu\text{m}^2$. Of course, when electron beam lithography is used, the device area of V-EGT can be further reduced.” (See page 31 in the Supplementary Materials, line 3-18)

Supplementary Fig. S20. Calculation of V-EGT device footprint in array configuration. a)

3D schematics of V-EGT. **b)** Top-down view of **a**.

Comment 3:

There is missing information in the experimental section: since there are 4 electrodes embedded into SiO₂, how are the connections established experimentally? Do they impact the scalability of this type of device? Figure 1c and Figure 1e only show the position of these connections.

Reply to Comment 3:

Thanks for the reviewer's comment. In the original manuscript's experimental section, we have provided an explanation of how to establish the connection between the TiN layers shown in TEM image (Fig. 1c) and the electrode pads located at different positions in the optical micrograph (Fig. 1e). We stated, "To facilitate subsequent electrical testing, each layer of TiN electrodes was exposed using a dry etching process, with each layer having different physical positions on the surface of the Si wafer." In the revised manuscript, we have provided a more detailed and explicit description of this process. A corresponding illustration (Fig. R5) is also added here for the reviewer's convenience.

Figure R5. Exposure of TiN located at different layers beneath SiO₂ in V-EGT. a. Optical micrograph of four-layer dual-gate V-EGT. **b.** Cross-sectional schematic of a four-layer dual-gate V-EGT along the white dashed line in **a**.

In the main text, we have added the following content:

“After etching the vertical sidewalls, to facilitate subsequent electrical testing, it was necessary to expose the TiN located beneath the SiO₂ at different levels. Initially, the eight horizontal electrode pad positions in **Fig. 1e** had the same height, consisting of a TiN/SiO₂/TiN/SiO₂/TiN/SiO₂/TiN/SiO₂ stack. To expose the first TiN, we etched away the top SiO₂ at the five rightmost electrode pad positions in **Fig. 1e**, thereby exposing the first TiN electrode. Next, at the third electrode pad position from the left, we sequentially etched away the SiO₂, TiN, and SiO₂, exposing the second TiN electrode. This process was repeated, and we successively exposed the third (at the second electrode pad position from the left) and the fourth (at the first electrode pad position from the left) TiN electrodes. With this procedure, all the TiN electrodes covered by the SiO₂ were exposed, enabling convenient measurement of the electrical characteristics of V-EGT devices.” (See page 23 in the main text, line 1-11 from the bottom and page 24 in the main text, line 1-2)

We have already pointed out in **Reply to Comment 2 of reviewer #2** that the device area of V-EGT in an array configuration is more meaningful compared to single-device scenario. According to the formula for calculating the device area of V-EGT in an array configuration, which is twice the source/drain electrode width multiplied by twice the gate electrode width, the electrode pad area of V-EGT and its associated device fabrication process have little effect on the scalability of V-EGT.

Comment 4:

Figure S7 is not mentioned in the main text. Furthermore, a short paragraph is required to describe properly these schemes.

Reply to Comment 4:

Thanks for the reviewer's comment. In the original manuscript, we did mention **Supplementary Fig. 7**, as stated below:

"Specifically, the retention of EGT depends on the discharge speed of EGT in the gate-source circuit after experiencing a gate pulse ($\tau_{RC} = R \times C$) from a circuit perspective. Here, R comprises the EGT electrolyte resistance and the external resistance in series with EGT, and C contains the gate and channel capacitance of EGT (Fig. 2d and Supplementary Fig. 7)⁵³." (See page 9 in the main text, line 6-10)

As recommended by the reviewer, in the revised manuscript's Supplementary Materials, we have provided an enriched figure caption for Supplementary Fig. 7 and added a paragraph to describe Supplementary Fig. 7. We have included the relevant information below for the reviewer's convenience.

“

Supplementary Fig. S7. The discharge circuit and corresponding equivalent circuit model of EGT. a) Schematic diagram of the experimental setup for EGT. **b)** Equivalent circuit diagram of the gate-source write circuit of EGT (indicated by the red dashed box in a. **c)** Simplified equivalent circuit diagram of the gate-source write circuit of EGT.

Supplementary Fig. S7a shows the schematic diagram of the experimental setup for EGT. It involves applying a varying gate voltage at the gate of EGT, applying a constant DC bias voltage (such as 0.1 V) at the drain, and grounding the source. An electronic

switch is connected in series between the gate of EGT and the pulse source.

Supplementary Fig. S7b presents the equivalent circuit diagram of the gate-source write circuit of EGT (indicated by the red dashed box in **Supplementary Fig. S7a**). In the circuit, R_{gate} , C_{gate} , R_{elec} , R_{ch} , C_{ch} , and V_{EGT} represent the gate/electrolyte interface resistance, gate/electrolyte interface capacitance, electrolyte/channel resistance, electrolyte/channel capacitance, and the voltage across the gate and source of EGT, respectively. Considering that the leakage current at both the gate/electrolyte interface and the electrolyte/channel interface is very small, R_{gate} and R_{ch} can be neglected. Therefore, **Supplementary Fig. S7b** can be simplified to **Supplementary Fig. S7c**.

During the interval between two adjacent gate pulses (i.e., the OFF pulse), the previously charged C_{gate} and C_{ch} discharge in the opposite direction of charging, as indicated by the direction of the discharging current, $I_{\text{discharge}}$. The retention of EGT can be quantified by the discharging time of the gate-source write circuit. Based on **Supplementary Fig. S7c**, the discharge time or speed of the gate-source write circuit can be expressed as $\tau_{\text{RC}} = R \times C$, where R includes the electrolyte resistance and the resistance of the electronic switch in series with the gate, and C contains C_{gate} and C_{ch} .” (See pages 9 and 10 in the Supplementary Materials)

Comment 5:

Figure 4d is not mentioned in the main text. Please add a corresponding sentence.

Reply to Comment 5:

Thanks for the reviewer's comment. In the revised manuscript, we have added the label “**Fig. 4d**” next to its content. We now provide below the modifications made in the revised manuscript for the reviewer's reference.

“To achieve this, a current threshold I_{th} is set, and the drain current is only larger than I_{th} when both gates act simultaneously (**Fig. 4d**).” (See page 16 in the main text, line 4-6)

Comment 6:

The following sentence “In this setup, the gate (drain) of a dual-gate V-EGT is considered as the dendrite (axon) of a neuron, and our device can mimic signal propagation pathways in the brain” should be rephrased, because it is not clear the role of the gate/drain. Gate and drain have two distinct roles in the device operation.

Reply to Comment 6:

Thanks for your comment. We agree with the reviewer's viewpoint that the gate and drain of EGT play distinct roles in the operation of the device. In EGT devices, the gate is responsible for controlling the magnitude of the channel current, while the drain is used for reading the magnitude of the channel current. In the mapping of our dual-gate V-EGT to biological neurons, the gate of the dual-gate V-EGT corresponds to the dendrite of a biological neuron, while the drain of the dual-gate V-EGT corresponds to the axon of a biological neuron.

In the revised manuscript, we have rephrased the sentence as follows:

"In this setup, with the gate of a dual-gate V-EGT being considered as the dendrite of a neuron and the drain of a dual-gate V-EGT as the axon of a neuron, our device can mimic signal propagation pathways in the brain." (See page 15 in the main text, line 1-3)

Comment from the Reviewer #3

The manuscript by Xu et al. describes a low-power synaptic transistor which can be used for dendritic computing. They address challenges in EGT fabrication by using a vertical channel which reduces the device footprint and facilitates fabrication dense arrays without added complexity/lithography steps. The devices are well characterized and show impressive low energy operation. The manuscript also includes a demonstration of devices used for coincidence detection which clearly demonstrates the utility of the presented devices. Overall, the work presents a significant advance in the development of synaptic transistors and warrants publication in Nature Communications after minor revisions. I have added a few technical comments that I

would like to see addressed before the paper can be accepted for publication.

Reply:

We sincerely appreciate the reviewer's positive evaluation for our work. These insightful comments and suggestions have significantly improved the quality of our paper. We have provided point-by-point response as follows.

Comment 1:

On line 42: the size of the human brain is compared to a football, but this size means different things depending where in the world the reader is from. I suggest also including a number for the rough volume of the brain.

Reply to Comment 1:

We thank the reviewer for this comment. In the revised manuscript, a numerical value regarding the approximate volume of the human brain has been added. The corresponding modification has been pasted below for the reviewer's convenience.

"The human brain possesses unparalleled cognitive capabilities, occupies a small footprint (roughly the size of a football; ~1200 - 1700 cm³, depending on individuals), and yet consumes very little power (approximately 20 W)." (See page 3 in the main text, line 2-4)

Comment 2:

On line 49: It would be helpful to clarify this motivation, as digital hardware does not necessarily "suffer" from the slowdown Moore's Law, instead digital hardware is not improving at an exponential pace anymore, limiting the ability to make increasingly complex ANNs without increasing compute times.

Reply to Comment 2:

We appreciate the reviewer's comment and acknowledge that our original manuscript did not provide a thorough explanation of this issue. In the revised manuscript, we have made corresponding modifications to address this concern, and the revised statement is pasted below for the reviewer's convenience.

"Nowadays, ANNs predominantly operate on digital hardware, which is not improving at an exponential pace anymore due to the slowdown of Moore's Law, limiting the ability to make increasingly complex ANNs without increasing compute times. Moreover, the von Neumann bottleneck, an inherent limitation of digital hardware, further hinders the execution efficiency of ANNs." (See page 3 in the main text, line 10-14)

Comment 3:

On line 82: the channel length of 30 nm is not a representative number for the device footprint since it is in the vertical direction. Is there a fundamental limitation to the process that can be quoted here? Is the footprint limited by the lithography? Or the directionality of the etching step? The resistivity of the TiN source/drain contacts or channel?

Reply to Comment 3:

We appreciate the reviewer's viewpoint that using the channel length of V-EGT to represent its device footprint is not rigorous enough. In the revised manuscript, we have modified this statement accordingly, and the revised version is provided below for the reviewer's convenience.

"In this study, we employed material and structure engineering to develop a vertical EGT (V-EGT) based neurotransistor that simultaneously exhibits STM, an ultra-short channel (30 nm), low energy consumption (read power ~ 3.16 fW, read energy ~ 30 fJ), and dual gates, making it an ideal choice for hardware implementation of dendritic computing, such as dendrite integration, digital and analog coincidence detection." (See page 4 in the main text, line 7-11 from the bottom)

Furthermore, regarding the calculation of the device footprint of V-EGT in an array configuration and its potential influencing factors, we have added relevant content in both the revised manuscript's main text and Supplementary Materials.

The added content in the main text is as follows:

“(see **Supplementary Note 3** for more discussion on the calculation of V-EGT device footprint in array configuration)” (See page 13 in the main text, line 12-14)

The added content in the Supplementary Materials is as follows:

“For V-EGT, the effective device area is considered to be the projection area of the core region on the horizontal plane. Therefore, the area of V-EGT is determined by the electrode width of the source/drain and the gate width. It can be calculated by the product of the source/drain electrode width and the gate width. Accurately, it should be twice the source/drain electrode width multiplied by twice the gate width, considering the spacing of one feature size between adjacent source/drain electrodes and adjacent gates (**Supplementary Fig. 20**).

The device area of V-EGT is limited by the lithographic precision which determines both the source/drain electrode width and the gate width. In our experiments, the typical source/drain electrode width is $10\ \mu\text{m}$, and the minimum gate width is $2\ \mu\text{m}$. Therefore, the smallest achieved area of our V-EGT is $10\ \mu\text{m} \times 2\ \mu\text{m} = 20\ \mu\text{m}^2$. Within the capability of ultraviolet lithography, the source/drain electrode width can be reduced to a minimum of $1\ \mu\text{m}$, and the gate width can also be reduced to a minimum of $1\ \mu\text{m}$. Hence, the minimum device area of our V-EGT can reach $1\ \mu\text{m} \times 1\ \mu\text{m} = 1\ \mu\text{m}^2$. Of course, when electron beam lithography is used, the device area of V-EGT can be further reduced.

Supplementary Fig. S20. Calculation of V-EGT device footprint in array configuration. a) 3D schematics of V-EGT. **b)** Top-down view of **a**.

From a manufacturing process perspective, the etching process still have some influence on the device area of V-EGT, as the width of the source/drain electrode, obtained through etching, is easier to achieve in a larger dimension (e.g., 10 μm) compared to a smaller dimension (e.g., 100 nm). Furthermore, the resistance of TiN source/drain electrodes is also a limiting factor in the device area of V-EGT due to the finite resistivity of TiN. The TiN source/drain electrodes must have a sufficient width to ensure proper device operation. This prerequisite restricts the source/drain electrode width from being arbitrarily small.” (See page 31 in the Supplementary Materials and page 32 in the Supplementary Materials, line 1-6)

Comment 4:

Is the long-term memory (shown in Fig. 2c,h,f) a permanent change in the channel conductance? Or is the transient very slow and will reach the original baseline conductance given enough time? If it is permanent, what is the difference in switching mechanism between the short-term and long-term memory processes?

Reply to Comment 4:

Thanks for the reviewer's comment. The long-term memory is not entirely a permanent change in the channel conductance. It contains a portion of permanent change and a significant portion of short-term memory. The latter is the transient and will reach the original baseline conductance given enough time.

As shown in **Fig. R6** (see **Supplementary Fig. S5** in the revised Supplementary Materials), EGT exhibits a hybrid mechanism of electric double layer and ion intercalation/deintercalation. The electric double layer mechanism arises from the accumulation of mobile electrolyte ions at the electrolyte/channel interface under the gate voltage. These ions diffuse back into the electrolyte after the voltage is removed, contributing to the short-term memory (transient part) of the electrical properties of EGT. In the ion intercalation/deintercalation mechanism, the mobile ions in the electrolyte are injected into the channel under gate voltage and require a reverse voltage

to remove them completely, contributing to the long-term memory (permanent part) of the electrical characteristics of EGT. At low excitation pulse intensities, the mobile ions inside the electrolyte lack sufficient energy to pass through the electrolyte/channel interface, leading to nearly complete transient electrical properties of the device. As the pulse excitation strength increases, a certain proportion of electrolyte ions can pass through the electrolyte/channel interface and enter into the channel, resulting in a hybrid mechanism of electric double layer and ion intercalation/deintercalation and a certain degree of permanent conductance change in the device.

Figure R6. Mechanism of EGT. The short-term memory characteristic of EGTs is due to their electric double layer operation mechanism, while the long-term memory characteristic of EGTs originates from their ion intercalation/deintercalation mechanism.

Comment 5:

The devices still operate quite slowly compared to other artificial synapses cited in the paper/SI. Is there a path towards pushing these materials to write time below 1 μs ? The projections in S14 are nice, but it seems it would be extremely challenging to scale electrolyte thickness down to the level necessary for μs operation.

Reply to Comment 5:

We agree with the reviewer's viewpoint that the current speed of our V-EGT devices is relatively slow. At present, the typical operating speed, that is, write time, is 1 ms, corresponding to a pulse voltage amplitude of $V_w = 3.5$ V. A higher gate voltage could lead to faster operation speed. We suggest that the V-EGTs are capable of achieving operation speeds in the sub-millisecond range at higher pulse amplitudes. Under a high gate voltage of 10 V, our V-EGT achieves a write pulse width of 10 μ s (**Fig. R7**). Nevertheless, it should be pointed out that such a method of reducing the write time comes at the cost of increasing the operating voltage of the device.

Figure R7. Current-time response of V-EGTs under gate pulse condition of $V_w=10$ V and $t_w=10$ μ s.

In the **Supplementary Note 2** of the original Supplementary Materials, we have discussed strategies for reducing the decay time of STM EGTs. In fact, the strategies proposed in that discussion are also applicable to improving their operation speed. If not preclude the possibility of adopting other materials, there are three methods that can be employed: (1) reducing the thickness of the electrolyte, (2) using an electrolyte material with higher ion conductivity, and (3) heating. The first method aims to improve the operation speed by reducing the distance traveled by electrolyte ions. The second and third methods accelerate the movement of electrolyte ions to achieve this goal.

Although electrolyte ions have significantly lower mobility in solid-state environment when compared with a liquid one, all-solid-state EGT devices can still achieve write times of less than 1 μ s when considering only the write pulse width of

EGT. For example, Tang *et al.* demonstrated write pulse widths as low as 5 ns using WO_x channel and LiPON electrolyte (Tang *et al.*, *IEDM*, 2018). Similarly, Lee *et al.* achieved a write pulse width of 10 μs using WO_x channel and AlO_x electrolyte (Lee *et al.*, *VLSI*, 2021). In our previous work, based on the same channel and electrolyte, the LTM EGT devices constructed have achieved sub-microsecond write pulse widths of 100 ns (Li *et al.*, *Adv. Mater.*, 2020). Therefore, it is foreseeable that the V-EGT devices will be capable of achieving sub-microsecond operation speeds ($< 1 \mu\text{s}$) in the near future.

In the Supplementary Materials, we have added the following content:

“The strategies proposed for accelerating the decay process of EGTs are also applicable to improving their operation speed partially. That is, if not preclude the possibility of adopting other materials, there are three methods that can be employed: (1) reducing the thickness of the electrolyte, (2) using an electrolyte material with higher ion conductivity, and (3) heating. Therefore, when these methods are adopted, it is foreseeable for V-EGTs to achieve sub-microsecond operation speeds ($< 1 \mu\text{s}$) in the near future.” (See page 30 in the Supplementary Materials, line 1-7)

Comment 6:

Since the capacitance of the electrolyte depends on the area of overlap, would the switching speed or the device scale with the channel width? Could the gate contact geometry be optimized to improve switching speeds? Or are there other strategies to increase speeds to $< 1 \mu\text{s}$?

Reply to Comment 6:

Thanks for the reviewer’s comment. We suggest that the gate width of EGT devices has little effect on the operation speed.

From a circuit perspective, we have pointed out that the discharge time constant of the gate-source circuit of EGT can serve as an indicator for EGT's retention characteristics. Since the write and discharge circuit of EGT is the same circuit, which is the gate-source circuit of EGT, the discharge time constant of the gate-source circuit

can also be used as an indicator for EGT's operation speed from a circuit perspective. However, unlike the gate-source circuit evaluated for retention characteristics, where the electronic switch in series with the gate of EGT is in the OFF state to achieve better retention, in the new circuit for the evaluation of EGT's operation speed, the electronic switch must be in the ON state because writing operations cannot be performed otherwise. In this case, as shown in **Fig. R8a**, the resistance in the gate-source RC circuit is no longer determined by the OFF-state resistance of the electronic switch but is instead determined by the electrolyte resistance. Under this condition, as shown in **Fig. R8b**, although the reduction in channel width will result in a decrease in capacitance, the resistance, specifically, the electrolyte resistance, will increase at the same rate. Therefore, the overall RC time constant of the circuit will remain unchanged, indicating that the operation speed of EGT will not decrease with a reduction in the EGT channel width. Therefore, based on the above analysis, we suggest that the operation speed of EGT devices does not vary with the EGT channel width, and optimizing the gate contact morphology cannot be used to optimize the operation speed of EGT devices.

Figure R8. a. Equivalent circuit diagram of the EGT gate-source circuit in the programming phase. **b.** The RC time constant of the circuit shown in **a** will not vary with changes in the gate width because although the capacitance in the circuit (gate capacitance and channel capacitance, C_{gate} and C_{ch}) will decrease with a reduction in the gate width, the resistance in the circuit (electrolyte resistance, R_{elec}) will increase at the same rate with a decrease in the gate width, thus keeping the overall RC time constant (τ_{RC}) unchanged.

Regarding the reviewer's third sub-question, "Or are there other strategies to increase speeds to $< 1 \mu\text{s}$?", we have already addressed it in **Reply to comment 5 of reviewer #3**. We now paste it again below for the convenience of the reviewer's review. "That is, if not preclude the possibility of adopting other materials, there are three methods that can be employed: (1) reducing the thickness of the electrolyte, (2) using an electrolyte material with higher ion conductivity, and (3) heating." (See pages 30 in the Supplementary Materials, line 2-5)

Comment 7:

Is there crosstalk between devices in the bilayer structure? Or is the read voltage so low that it does not disturb the resistance state of the adjacent channel?

Reply to Comment 7:

Thank you for the comment. As the reviewer pointed out, there indeed exist crosstalk between two single-gate V-EGTs when regarding a dual-gate V-EGT as two single-gate V-EGTs with shared source/drain electrodes. If one of the single-gate V-EGT is read, the channel resistance of the other single-gate V-EGT will also be passively read. However, as the reviewer mentioned, the crosstalk effect is minimal due to the extremely low read voltage (usually around 0.1 V, and in our work, it can even be as low as 0.1 mV).

REVIEWERS' COMMENTS

Reviewer #1 (Remarks to the Author):

All the concerns are satisfactorily addressed. I have no objection to publishing this manuscript.

Reviewer #3 (Remarks to the Author):

The authors' revisions have greatly clarified the technical details of the paper. I therefore recommend the article for publication Nature Communications without further revisions.

We sincerely appreciate the editor's time in handling our manuscript and are delighted that reviewers are positive about our work. We are grateful to the reviewers for their valuable comments and have addressed their comments point-by-point as listed below.

Comment from the Reviewer #1

All the concerns are satisfactorily addressed. I have no objection to publishing this manuscript.

Comment from the Reviewer #2

The authors' revisions have greatly clarified the technical details of the paper. I therefore recommend the article for publication Nature Communications without further revisions.

Reply:

We thank both referees for the valuable comments and constructive suggestions throughout the revision, which significantly improves the quality and clarity of the manuscript.